# The Crosstalk between Mesenchymal Stromal/Stem Cells and Hepatocytes in Homeostasis and under Stress

**DOI:** 10.3390/ijms242015212

**Published:** 2023-10-16

**Authors:** Irina V. Kholodenko, Roman V. Kholodenko, Konstantin N. Yarygin

**Affiliations:** 1Laboratory of Cell Biology, Orekhovich Institute of Biomedical Chemistry, 119121 Moscow, Russia; 2Laboratory of Molecular Immunology, Shemyakin-Ovchinnikov Institute of Bioorganic Chemistry, Russian Academy of Sciences, 117997 Moscow, Russia; khol@mail.ru

**Keywords:** mesenchymal stem cells, hepatocytes, liver diseases, cell-to-cell communication, regeneration, apoptosis, ferroptosis, pyroptosis, lipotoxicity, mitochondrial transfer

## Abstract

Liver diseases, characterized by high morbidity and mortality, represent a substantial medical problem globally. The current therapeutic approaches are mainly aimed at reducing symptoms and slowing down the progression of the diseases. Organ transplantation remains the only effective treatment method in cases of severe liver pathology. In this regard, the development of new effective approaches aimed at stimulating liver regeneration, both by activation of the organ’s own resources or by different therapeutic agents that trigger regeneration, does not cease to be relevant. To date, many systematic reviews and meta-analyses have been published confirming the effectiveness of mesenchymal stromal cell (MSC) transplantation in the treatment of liver diseases of various severities and etiologies. However, despite the successful use of MSCs in clinical practice and the promising therapeutic results in animal models of liver diseases, the mechanisms of their protective and regenerative action remain poorly understood. Specifically, data about the molecular agents produced by these cells and mediating their therapeutic action are fragmentary and often contradictory. Since MSCs or MSC-like cells are found in all tissues and organs, it is likely that many key intercellular interactions within the tissue niches are dependent on MSCs. In this context, it is essential to understand the mechanisms underlying communication between MSCs and differentiated parenchymal cells of each particular tissue. This is important both from the perspective of basic science and for the development of therapeutic approaches involving the modulation of the activity of resident MSCs. With regard to the liver, the research is concentrated on the intercommunication between MSCs and hepatocytes under normal conditions and during the development of the pathological process. The goals of this review were to identify the key factors mediating the crosstalk between MSCs and hepatocytes and determine the possible mechanisms of interaction of the two cell types under normal and stressful conditions. The analysis of the hepatocyte–MSC interaction showed that MSCs carry out chaperone-like functions, including the synthesis of the supportive extracellular matrix proteins; prevention of apoptosis, pyroptosis, and ferroptosis; support of regeneration; elimination of lipotoxicity and ER stress; promotion of antioxidant effects; and donation of mitochondria. The underlying mechanisms suggest very close interdependence, including even direct cytoplasm and organelle exchange.

## 1. Introduction

Liver diseases are a global problem due to high morbidity and mortality. Europe has the highest number of cases of diagnosed liver disease in the world [1]; in the EU alone, nearly 30 million people live with chronic liver diseases [2]. Liver cirrhosis kills about 170,000 people every year, accounting for 1.8% of all deaths reported in Europe [2]. Liver diseases are now the second leading cause of disability in Europe after ischemic heart disease [3]. According to the Global Burden of Disease (GBD) Study 2017, the number of deaths from cirrhosis and other chronic liver diseases was 1.32 million in 2017 compared with 889,000 in 1990 [4]. Presumably, the COVID-19 pandemic had a pronounced negative impact on mortality from chronic liver diseases, especially exacerbating the situation in countries where the alcoholic etiology was predominant [5].

The current therapeutic approaches to the treatment of liver diseases are mainly focused on reducing the symptoms and slowing down the progression of the disease, while heterologous organ transplantation remains the only way to treat end-stage liver disease. In this regard, issues related to the development of new effective approaches aimed at stimulating liver regeneration, both by switching on the internal resources of the organism and by enhancing regeneration with drugs and various medicinal procedures, are of high relevance. In the field of regenerative medicine in general and in regenerative hepatology in particular, stem and progenitor cells are considered powerful therapeutic tools [6]. It has been shown that systemic infusion of MSCs effectively induces immunological tolerance [7] and immunosuppression [8] in liver transplant recipients. To date, several systematic reviews and meta-analyses have been published confirming the efficacy of mesenchymal stem cell (MSC) transplantation in the treatment of liver diseases of various severities and etiologies [9,10,11]. However, despite the successful use of MSCs in clinical practice and the promising therapeutic results that have been demonstrated in animal models of liver diseases, the mechanisms of their protective and regenerative action remain poorly understood. The data about the mediators of intercellular communication produced by these cells and responsible for their therapeutic action are fragmented and often contradictory.

MSCs are a specific cell type present in all tissues and organs of the body and participate in the formation of the organs’ stroma and tissue-specific niches. The unique features of these cells include immunomodulatory properties and extensive regenerative potential. As shown by several research groups, the liver contains a pool of resident MSCs, which, in terms of their main characteristics (expression patterns of cell surface markers, adhesion, and morphology), correspond to those of MSCs isolated from other tissue sources [12,13,14]. Liver MSCs have several unique characteristics compared with those obtained from other tissue sources. In particular, liver MSCs express a spectrum of liver-specific genes [15]. Lee et al. [16] showed that liver MSCs are more readily differentiated into the cells of hepatic lineage compared with umbilical cord-derived MSCs, while Yigitbilek et al. [17] demonstrated their more pronounced immunomodulatory properties and pro-angiogenic potential compared with adipose tissue-derived MSCs. As shown by Raicevic et al. [18], inflammatory conditions significantly influenced the immunophenotype of liver MSCs; the expression of the costimulatory molecule CD40 was initiated, the expression of Toll-like receptors (TLRs) 1, 2, and 3 increased, and the expression of the adhesion molecules CD58 and CD106 as well as the expression of the immunoregulatory molecule CD274 (PD-L1 or B7-H1) were substantially enhanced [18]. Girousse et al.’s review article [19] proposed and substantiated a hypothesis claiming that endogenous MSCs can serve as the main cellular components of an interorgan communication network ensuring coordinated functioning of the tissues, organs, and their parts at the cellular level and cooperative responses to stressful stimuli. Originally, the known interorgan communication messages included peptides, proteins, and metabolites, and later, the list was supplemented with extracellular vesicles. Since MSCs or MSC-like cells are present in all tissues and organs, it is likely that most of the key cell-to-cell interaction-dependent processes, including crosstalk between cells, the epithelial–mesenchymal transition, immunomodulation, differentiation, cellular maintenance, and cell death, are dependent on the MSCs present within the tissue niche. In this context, it is important to understand the mechanisms underlying the communication between MSCs and differentiated somatic cells in each particular tissue. This is important both from the perspective of basic science and for the development of therapeutic approaches involving the modulation of the activity of resident MSCs. With regard to the liver, the research should be concentrated on the intercommunication between MSCs and hepatocytes. Elsewhere, we discussed resident liver MSCs different from hepatic stellate cells and reviewed existing data concerning their possible triple origin from the resident liver MSCs formed during histogenesis and organogenesis, hepatic pericytes, and MSCs mobilized from the bone marrow [15]. The latter option is evident in situations of liver injury, such as liver fibrosis [20,21], liver cirrhosis [22,23], partial hepatectomy [24,25], and hepatocellular carcinoma [26], and after living donor liver transplantation [27]. The goal of this review was to identify the main pathways and key factors mediating the crosstalk between MSCs and hepatocytes to determine the possible mechanisms of the direct and indirect interaction of the two cell types under normal or close to normal conditions and under stress. Mainly, we focused on model experiments in vitro since in such artificial systems, the key regulators of intercellular communication are the most reliably detected, while in vivo, it is difficult to determine the true regulators due to the presence of many cell types and other biologically active components in the microenvironment.

## 2. Crosstalk between MSCs and Hepatocytes in Close-to-Homeostasis Conditions

### 2.1. Chaperon-Like Functions of MSCs in Mixed 3D Cultures

In liver organogenesis, mesenchymal cells play an important role in the development of the liver bud and the subsequent morphogenesis of the organ. The formation of the liver from the cell condensate begins with the loss of contact between hepatoblasts due to the suppression of E-cadherin expression, which provides contact between neighboring cells [28,29]. This allows hepatoblasts to migrate into the surrounding mesenchyme. Simultaneously, matrix metalloproteinases (MMPs) degrade extracellular matrix proteins, such as collagen, fibronectin, and basement membrane laminin, thereby facilitating hepatoblast migration. This extracellular matrix remodeling is mainly mediated by MMP2 expressed by mesenchymal cells and MMP14 expressed by hepatoblasts [30]. Matsumoto et al. [31] demonstrated that during embryonic development, mesenchymal cells play a critical role in the formation of the liver primordium from the endoderm, controlling hepatic progenitor cells and regulating their fate by creating a specialized hepatocyte embryonic niche.

The fetal liver is a unique tissue in terms of the coexistence of cells of endodermal and mesenchymal origin [32]. Ito et al. [33] showed that in the fetal liver, PDGFRa-positive mesenchymal cells are located in close proximity to albumin-positive hepatoblasts. In vitro, Dlk1^mid^PDGFRa^+^ mesenchymal cells secreted factors that increased hepatoblast proliferation. At the same time, direct interaction between hepatoblasts and Dlk1^mid^PDGFRa^+^ mesenchymal cells did not increase proliferation, but it strictly induced the differentiation of hepatoblasts into hepatocytes, as shown by the increased expression of genes, reflecting mature hepatocyte function; these genes included those encoding for albumin, tyrosine aminotransferase (TAT), and carbamoyl-phosphate synthetase (CPS) [33]. In vitro mouse fetal CD49f^±^Thy1^+^gp38^+^CD45^−^ mesenchymal cells cocultured with mouse fetal hepatoblasts induced their maturation, as shown by positive acid–Schiff staining and the upregulation of the expression of mature hepatocyte markers, such as tyrosine aminotransferase, tryptophan-2,3-dioxygenase, and glucose-6-phosphatase [34]. Also, human bone marrow MSC-derived conditioned medium induced the maturation of hepatoblasts derived from induced pluripotent stem cells, as shown by an increase in the production of albumin and urea as well as an increase in the expression and activity of cytochrome P450 (CYP) 3A4 [35].

Many approaches to obtaining liver organoids (sometimes called liver buds) of diverse complexity based on combinations of various cell types have been developed, both to study the mechanisms of liver organogenesis and to obtain functionally active hepatocyte-containing tissue engineering constructs suitable for drug screening with the potential for clinical application [36,37]. Direct cell-to-cell contact, the interaction of cells with extracellular matrix proteins, and the presence of a whole range of soluble factors are critical for the successful formation of liver organoids [38]. In organoids consisting of monocultures of hepatocytes, hepatoblasts, and/or hepatocyte-like cells differentiated from various progenitor and stem cells matrix is a critical component required for the efficient assembly and maintenance of the functional activity. Collagen or matrigel is often used to provide a 3D environment for cells to interact and self-organize. More complex bioengineering approaches, such as the employment of decellularized organ scaffolds or bioprinting technology, can also be applied to generate a native 3D tissue structure. Along with the matrix, soluble factors imitating events occurring in organogenesis can be utilized; among them are a variety of small molecules and biologicals, such as Wnt agonists R-spondin1 and CHIR99021; EGF, FGF7, and FGF10; HGF and the TGF-β inhibitor A83-01; TNFα; etc. [39,40,41]. In liver organoids consisting of hepatocytes, endothelial cells, and mesenchymal cells, the supportive mesenchymal and endothelial cells play a pivotal role in maintaining the viability and functionality of the parenchymal cells. They provide hepatocytes with an appropriate microenvironment in the form of cell contact, the production of extracellular matrix proteins, vasculature, and the secretion of a whole range of soluble factors.

MSCs provide reinforcement of the 3D architecture of liver organoids. It has been established that organoids are not formed in the absence of stromal components, such as MSCs and/or extracellular matrix [37,42]. In particular, it has been shown that 3D spheroids derived from human BM-MSCs and human hepatocytes form an inner core consisting of parenchymal cells that express HNF4α and cytokeratin 18 surrounded by an outer layer of vimentin-expressing stromal cells. Phalloidin staining showed the presence of a peripheral actin belt and F-actin enrichment in the apical region of the cells, indicating hepatocyte polarization. Type I collagen deposits were observed in the intercellular space between mesenchymal cells and the internal parenchymal mass [43].

The formation of liver organoids in the presence of MSCs occurs due to the so-called “contraction mechanism”, which is non-muscular myosin-II (NSM-II)-dependent [36]. Using live cell tracking, Takebe et al. [44] showed that the co-cultivation of iPSC-derived hepatic endodermal cells, HUVECs, and MSCs initiated the assembly of vascularized organ buds after rapid cell convergence. This was followed by spatial rearrangements through self-organization, as evidenced by the formation of an endothelium-like network. During the initial phase of self-convergence, a cohesive multicellular unit was formed consisting of cells that quickly moved to a single center, a process called collective condensation. It is noteworthy that the formation of cell condensates was based not on the active migration of cells but on their contraction. Importantly, cellular condensation did not occur in the absence of MSCs, which indicates that MSCs provide sufficient contractile force for the assembly of the three cell types. During embryonic invagination, a group of cells undergoes contraction. A sharp inward shift of intercellular junctions occurs due to the activity of myosin II (MII), which allows cells to invaginate during embryonic gastrulation [45,46]. MII regulatory light chain diphosphorylation (ppMRLC) has greater actin-activated Mg^2+^-ATPase activity than MRLC monophosphorylation (pMRLC), which is sufficient to generate contractile force [47]. It was shown that the level of ppMRLC in co-cultures of iPSC-derived hepatic endodermal cells, HUVECs, and MSCs significantly increased after 4 h of plating, and a marked increase in the total amount of MRLC was observed during the formation of cell condensates, which corresponded to the time when the cells began their collective movements. MIIA immunostaining in the liver bud showed that CD90^+^ MSCs significantly expressed MIIA. These results clearly indicate that MSCs in the liver bud provide the traction force created by the actomyosin complex of the cytoskeleton, which plays an important role in directed cell movements and the formation of cell condensates [44]. Later, the same group of authors developed a system for obtaining liver buds from three types of cells, each of which was derived from human induced pluripotent stem cells (iPSC) by directed differentiation into transitional hepatic endoderm (tHE) cells (co-expressing T-box transcription factor 3 (TBX3) and adrenoceptor alpha 1B (ADRA1B)), septum transversum mesenchyme (STM) cells (co-expressing Wilms tumor 1 homolog (WT1), MIIA, and LHX2), and endothelial progenitors (iPSC-EC) (co-expressing CD144 (VE-cadherin) and CD31), respectively [48]. Global gene array analysis showed that the iPSC-STM signatures were similar to those of human adult hepatic mesenchymal cell signatures [49,50]. In the liver bud, these cells were the driving force of myosin-IIA-dependent self-condensation [48].

Liver organoids generated using decellularized liver tissue as a scaffold made of the native extracellular matrix containing all ECM proteins essential for the maintenance of hepatocytes, such as collagen type I, collagen type IV, fibronectin, and laminin, represent one of the most practical in vitro models for hepatology research [36,51]. It has been shown that MSCs present in liver organoids increase the level of extracellular matrix proteins and enhance the expression of β-integrin by hepatocytes, securing their effective adhesion to the ECM [51]. Also, MSCs, being the cells that form tissue niches for various types of stem cells, are themselves efficient producers of ECM proteins, including laminin, fibronectin, type I collagen, tenascin-C, decorin, etc. [52,53]. Accordingly, extracellular matrix proteins produced by MSCs provide additional supportive substrates for hepatocytes within liver organoids, as demonstrated by Park et al. [42].

Several studies have shown that liver organoids fail to assemble if MSCs are replaced with an MSC-derived conditioned medium (MSC-CM) [44,54], although the paracrine effects of MSCs may still persist, supporting the differentiation and maturation of hepatocytes [54,55]. For example, paracrine factors, including IL-6, GRO-α, IL-8, MCP1, SDF-1α, LIF, VEGF-A, and PDGF-BB, secreted by human amniotic-derived MSCs (hAMSCs) cultured under 3D conditions had a pronounced effect on hepatogenic differentiation and the functional maturation of EpCAM^+^ liver progenitor cells in a monoculture organoid, as evidenced by the over-production of albumin and the over-activity of CYP3A4 [55]. The involvement of MSCs in niche formation in liver organoids may not be limited to cell-to-cell contact, paracrine signals, and the production of extracellular matrix proteins, but it may also involve the differentiation of MSCs into endothelial cells with subsequent formation of vascular structures. Kadota et al. [51] demonstrated the co-expression of VEGF and CD31 on BM-MSCs in liver organoids, suggesting that MSCs can be differentiated in endothelial lineage cells, probably through interactions between co-cultured hepatocytes and the microenvironment of the liver post-decellularized ECM. These data demonstrate that under 3D conditions, MSCs play a key role in the creation and maintenance of the tissue architecture of liver organoids, providing niche formation and paracrine signals directed at parenchymal cells. Schematically, the role and functions of MSCs in the liver organoids are shown in Figure 1.

Beneficial effects of MSCs on cultured hepatocytes were observed not only in autologous/syngeneic and allogeneic systems but also in xenogeneic systems. Montanari et al. [56] showed that human bone marrow MSCs co-encapsulated with porcine hepatocytes in alginate- and poly(ethylene glycol) (PEG)-grafted alginate hydrogels significantly prolonged hepatocyte survival and stimulated albumin production without affecting diazepam metabolism [56]. Organoids generated by co-cultivation of primary rat hepatocytes with human umbilical cord-derived MSCs in porcine liver-derived ECM (PLECM gel) survived longer than those made of hepatocytes only, maintained hepatocyte function for more than 20 days, showed a significantly higher production of albumin on day 14, and demonstrated enhanced urea synthesis for more than 3 weeks [36]. On the other hand, in 3D spheroids derived from human BM-MSCs and human hepatocytes, the levels of urea production and albumin secretion did not change compared with those of monocultures of hepatocytes [43]. However, despite the fact that the expression of CYP3A4 increased both in the 3D co-culture of hepatocytes with BM-MSCs and in the monoculture of hepatocytes, the activity of this enzyme persisted for 2 weeks only in the co-culture, while in the monoculture, it decreased after a week. Also, when hepatocytes were co-cultured with BM-MSCs, the expression of the 1A2 and 2C9 isoforms significantly increased compared with monoculture [43], suggesting improved maintenance or activation of the metabolic function of hepatocytes due to the presence of MSCs.

### 2.2. Crosstalk of Hepatocytes and MSC in 2D Co-Culture

Most papers describing simpler, essentially 2D systems of hepatocyte co-cultivation with MSCs have also demonstrated the supporting role of the latter, manifested by the increased viability of hepatocytes as well as better preservation of their typical morphology and other signs of a differentiated state, such as albumin and urea production and enhanced drug metabolism [57].

Using an allogeneic direct co-culture system in which human primary hepatocytes were seeded on top of a human adipose-derived MSC monolayer, Qin et al. [58] showed that, in addition to providing hepatocyte support, MSCs had pronounced hepatotrophic and anti-apoptotic effects. Minimal hepatotrophic and anti-apoptotic effects were also shown in indirect co-cultures of hepatocytes with MSCs in the transwell plates and could be induced by conditioned media. Co-cultures with MSCs reduced the secretion of TNF-α by hepatocytes, and this inhibitory effect of co-cultures on the hepatocyte autocrine depended on MSC–hepatocyte contact and the intracellular ROS activity in the MSCs. TNF-α is known to have a bidirectional regulatory effect on both hepatocyte proliferation and apoptosis [59]. Co-cultures with MSCs protect hepatocytes from apoptosis induced by high levels of TNF-α [60]. Co-culturing hepatocytes with MSCs also induced an additive or synergistic contact-dependent effect on TGF-β1 production. The autocrine action of TGF-β1 on MSCs, leading to increased production of ECM proteins [61], was required for the trophic effects seen in co-culture conditions. Direct co-culturing significantly increased TGF-β1 secretion compared with monocultures of hepatocytes or MSCs, while in indirect co-cultures, these effects did not take place [58]. Extracellular collagen deposition by MSCs was enhanced by co-cultivation with hepatocytes, and this at least partially contributed to the trophic and anti-apoptotic effects of MSCs. Attachment to extracellular collagen facilitated the entry of hepatocytes into the S phase [62] and mediated hepatocyte aggregation and cell-to-cell contact [63]. Extracellular collagen is mainly localized around MSCs, and the inhibition of the production of type I collagen in MSCs significantly reduces the synthesis of albumin and urea by hepatocytes [64]. Increased deposition of extracellular collagen by mesenchymal cells may predominantly occur due to MSC–hepatocyte contact and is associated with intracellular ROS activity in MSCs. These results suggest that the hepatotrophic effect of co-culturing is mostly mediated by the synergistic effects of ECM and direct cell-to-cell interaction [58].

MSCs isolated from different tissue sources have slightly different effects on hepatocytes in co-cultures. This is probably due to differences in the expression/secretion profiles of hepatotrophic factors. For example, under standard monoculture conditions, human umbilical cord perivascular cells (HUCPVCs) have been shown to express lower levels of hepatotrophic genes, such as cytokine stem cell factor (KITLG), proteoglycan decorin (DCN), notch ligand jagged 1 (JAG1), and HGF, compared with human BM-MSCs. At the same time, HUCPVCs and BM-MSCs equally supported the production and secretion of albumin as well as the expression of the hepatospecific protein tryptophan 2,3-dioxygenase, although the expression of tyrosine aminotransferase was upregulated only in the co-cultures of hepatocytes with HUCPVCs. Unlike BM-MSCs, HUCPVCs effectively supported ureagenesis. BM-MSCs, in turn, were more effective at increasing CYP activity [65]. Figure 2 presents a scheme of the mutual influence of hepatocytes and MSCs during co-cultivation under normal conditions.

It is noteworthy that in addition to the effects of MSCs on hepatocytes, the latter also affect mesenchymal cells. In human HUCPVCs and BM-MSCs, the expression profiles of hepatotrophic factors significantly changed when they were co-cultured with rat hepatocytes. HUCPVCs began to express increased levels of laminin (LAMB1), HGF, IL-6, and connexin 43 (GJA1) compared with BM-MSCs, and BM-MSCs expressed increased levels of KITLG and JAG1 [65]. Porcine hepatocytes induced increased expression and secretion of serpin E1 (plasminogen activator inhibitor-1 (PAI-1)) by human MSCs and did not affect the secretion of pro-inflammatory cytokines such as CCL2, CXCL12, and macrophage migration inhibitory factor (MIF) [56]. Notably, several studies have shown that MSCs do not differentiate in the hepatocyte lineage in co-cultures [65,66].

In the majority of publications describing the direct co-cultivation of hepatocytes and MSCs, it has been shown that cell-to-cell contact is necessary for the implementation of maintenance effects, although paracrine interactions between cells due to the production of bioactive molecules may play a significant role. In several works described above, it was shown that an MSC-CM had no or only partial supporting and/or protective effects on hepatocytes [58,67,68]. For example, a conditioned medium derived from human adipose MSCs stimulated the proliferation of mouse hepatocytes and the accumulation of glycogen but did not affect the activity levels of cytochrome P450 or the production of albumin and urea [67]. Similar results were obtained with regard to the action of the conditioned medium obtained from human MSCs on human hepatocytes, namely improved survival of hepatocytes and a slight increase in their functional activity compared with direct co-cultures [68]. The mechanisms of MSC communication with target cells mediated by cell-free MSC derivatives, including extracellular vesicles, conditioned media, apoptotic bodies, and mitochondria, have been extensively characterized in various disease models [69].

## 3. MSCs Communication with Hepatocytes in Stressful Environment

Generally, the communication of MSCs with hepatocytes as well as with other cell types can be carried out not only through direct interaction between cells but also through the secretion of soluble factors [70,71,72], the exchange of extracellular vesicles [73,74], or mitochondrial transfer [75]. For example, mouse BM-MSC-derived exosomes induced the conversion of mature hepatocytes into EpCAM^high^ small oval cells. These cells, which are hepatic progenitor cells, also expressed hepatoblast markers, including DLK1, Onecut 2 (OC2), α-fetoprotein, and cytokeratin 19 [76]. On the other hand, the MSC-CM had a supportive effect on hepatocytes, stimulating their metabolism and albumin production [67]. A human amnion-derived MSC (hAMSC)-CM had the potential to increase hepatic stem/progenitor cell differentiation in human liver organoids [55].

The following two sections present the results of the in vitro and in vivo studies demonstrating the protective effects of both MSCs and MSC derivatives on hepatocytes exposed to stress in the context of the crosstalk between the two cell types. The mechanisms involved in hepatocyte protection against apoptosis, ferroptosis, pyroptosis, and lipotoxicity are described in detail.

### 3.1. Anti-Apoptotic Effects of MSCs

Apoptosis is a ubiquitous form of cell death that regularly occurs in human liver diseases. Under normal physiological conditions and during organ development, apoptosis is carefully regulated spatiotemporally and does not provoke adverse secondary events. Pathological apoptosis, on the contrary, activates secondary signaling cascades that can lead to massive damage to tissues and organs. Such pathological apoptosis-induced cascades stimulate tissue inflammatory responses, injury, and fibrosis [77]. Apoptosis is a trigger for both acute and chronic liver diseases. For example, the number of apoptotic hepatocytes increases significantly in patients with non-alcoholic steatohepatitis and correlates with the severity of the disease and the stage of fibrosis, which indicates the etiopathogenetic role of apoptotic cell death in the progression of this disease [78,79]. Infection with hepatitis B (HBV) or C viruses (HCV) induces liver damage, mainly by activating the host’s immune response to viral proteins expressed by infected hepatocytes. Notably, acute liver injury is mainly characterized by hepatocyte necrosis, while chronic HCV and HBV infections predominantly induce massive apoptosis. In viral hepatitis, cytotoxic T lymphocytes recognize viral antigens on hepatocytes and kill them, leading to liver damage. It has been proven that cytotoxic T lymphocytes kill virus-infected hepatocytes through Fas-dependent apoptosis. Consequently, hepatocyte Fas expression is significantly increased, and the number of FasL-positive infiltrating mononuclear cells in the liver is significantly elevated in patients with hepatitis C [80] or chronic active hepatitis B [81]. Fas/FasL expression levels are closely correlated with inflammatory activity, which initiates the development of the disease and contributes to its progression [82]. Another mechanism by which virus-infected hepatocytes undergo massive apoptosis involves TNF-α [83,84]. Long-term alcohol consumption in large quantities leads to the development of alcoholic liver disease [85]. Alcohol induces the production of high levels of reactive oxygen species (ROS) and a decrease in intracellular antioxidant levels, thereby causing oxidative stress in hepatocytes. Alcohol-induced oxidative stress is the cause of massive hepatocellular apoptosis and the main mechanism leading to liver damage [86]. The mechanisms of hepatocyte apoptosis in various pathological liver conditions have been described in more detail in a number of reviews [77,82,87].

#### 3.1.1. Involvement of Growth Factors and Bioactive Molecules in the Anti-Apoptotic Effects of MSCs

Many authors have noted decreased apoptosis and increased proliferation in hepatocytes after the transplantation of MSCs or cell-free MSC derivatives in various animal models of liver diseases [88,89,90,91]. For example, BM-MSC-derived exosomes entered primary rat hepatocytes within 12–24 h and accumulated in the cytoplasm, preventing hepatocyte apoptosis induced by D-galactosamine and lipopolysaccharide (D-GalN/LPS) by increasing the number of autophagosomes and the expression of the autophagy-related proteins LC3II and Beclin-1. At the same time, the expression levels of the proapoptotic proteins Bax and cleaved caspase 3 significantly decreased and the expression level of the anti-apoptotic protein Bcl-2 increased, indicating the induction of autophagy in hepatocytes due to exosomes preventing apoptosis [92]. Exosomes prepared from hESC-derived HuES9.E1 MSCs stimulated hepatocyte proliferation and reduced the apoptosis of hepatocytes in a mouse carbon tetrachloride liver injury model. In two in vitro models, the H_2_O_2_-induced oxidative stress model and the acetaminophen-induced injury model, it was shown that in three lines of mouse and human hepatocytes, exosomes derived from the hESC-derived HuES9.E1 MSCs regulated hepatocyte proliferation, promoting their transition from the G0 phase to the G1 phase of the cell cycle by stimulating the expression of tumor necrosis factor alpha (TNF-α), interleukin 6 (IL-6), inducible nitric oxide synthase (iNOS), cyclooxygenase-2 (COX-2), and macrophage inflammatory protein 2 (MIP-2), which in turn led to the restoration of NF-κB and STAT3 signaling. The anti-apoptotic effect of exosomes was associated mainly with the activation of STAT3 signaling, the inhibition of caspase 3, and increased levels of the anti-apoptotic protein Bcl-xL, but it was not associated with the activation of antioxidative genes, including heme oxygenase-1 (HO-1), glutathione peroxidase 4 (Gpx4), glutathione reductase (GSR), and MnSOD [93]. The level of apoptosis decreased and proliferation increased due to the activation of Notch signaling in direct co-cultures of mouse hepatocytes (H2.35) treated with palmitic acid to induce lipotoxicity and human adipose-derived MSCs [94].

Also, in vitro studies have shown that MSCs suppressed staurosporine-induced apoptosis in hepatocyte co-cultures [58]. Staurosporine is known to activate caspase 3 signaling independently from caspases 8, 9, and 12 [95]. Caspase 9 is an initiator of caspase-mediated apoptosis through the mitochondrial pathway [96]. Co-culture of hepatocytes with MSCs reduced caspase 9 gene expression rather than that of caspases 3, 8, and 12, suggesting that in the co-culture, the anti-apoptotic action of MSCs is mainly confined to the initiation of apoptosis. Bax/Bcl-2 balance regulates hepatocyte apoptosis [97]. BID interacts with Bax and mediates caspases 3 and 8 in apoptotic hepatocytes [98]. Co-cultures with MSCs significantly reduced the Bax/Bcl-2 ratio and BID expression by hepatocytes in a manner dependent on cell contact and MSC ROS activity. The authors suggested that isolated primary human hepatocytes became irresponsive to soluble trophic and anti-apoptotic factors released by MSCs, which was the reason for the weaker anti-apoptotic activity of the MSC-CM [58].

In an in vitro model, rat hepatocytes treated with sera from animals given D-galactosamine to induce acute liver failure underwent massive cell death and began to express high levels of IL-6. When such hepatocytes were co-cultured with human MSCs, their viability was restored, and IL-6 secretion was significantly reduced [66]. In this context, IL-6 expression can be regarded as a response to inflammatory conditions, enhancing hepatocyte damage. On the other hand, in an earlier work by Isoda et al., hepatocytes co-cultured with BM-MSCs maintained higher levels of albumin and urea production compared with hepatocyte monocultures, regardless of intercellular contact; this effect was mainly associated with IL-6 produced by BM-MSCs but not by rat hepatocytes [99]. IL-6 was significantly elevated in direct co-cultures of rat hepatocytes treated with the serum of an acute liver failure patient and human orbital fat-derived stem cells and was the key factor mediating the protective effects of MSCs [100]. Such contradictory results regarding the contribution of IL-6 to the protective effects of MSCs on hepatocytes may be related to the multiplicity of the mechanisms of IL-6 signaling.

IL-6 is known to have a cytoprotective effect in vivo and in vitro [101,102]. IL-6-treated hepatocytes are able to maintain higher levels of albumin and urea secretion and show a better ability to metabolize drugs in the presence of injurious agents [101]. IL-6 has an effect on liver regeneration and can induce hepatocyte proliferation, but in some cases, it can lead to the arrest of cell growth [103,104]. This may be explained by the results of Sun et al., who showed that IL-6 alone induced a resting state in hepatocytes but that the effect was reversed in the presence of non-parenchymal cells. IL-6 stimulates non-parenchymal cells to produce hepatocyte growth factor, which is a potent hepatic mitogen and induces hepatocyte proliferation [104].

The pleiotropic effects of IL-6 on hepatocytes may be associated with the specific regulation of its signaling [105]. IL-6 has an affinity to IL-6R but not to gp130. In turn, IL-6R also does not have a significant affinity to gp130. Only the formed IL-6/IL-6R complex can bind to gp130 [106]. This is a very important sequence of biologically active molecule interactions that plays a significant role in the regulation of IL-6 signaling. While gp130 is expressed by all body cells, IL-6R is expressed by only a few cell types, including hepatocytes, some leukocytes, and some epithelial (biliary epithelial cells) and non-epithelial cells (hepatic stellate cells). Therefore, only cells expressing IL-6R can directly respond to cytokine IL-6 [106]. Other cell types may respond to IL-6 by trans-signaling when the cytokine interacts with the soluble form sIL-6R. In this case, the IL-6/sIL-6R complex interacts with gp130 with high affinity. Since hepatocytes express more gp130 than IL-6R, the presence of IL-6 and sIL-6R results in greater gp130 activation and, therefore, higher IL-6 signal amplitude. Moreover, the IL-6/sIL-6R complex was shown to be internalized less efficiently than IL-6/IL-6R, resulting in longer IL-6 signal duration when mediated by trans-signaling [107]. Accordingly, hepatocytes permanently proliferated in IL-6/sIL-6R double transgenic mice in the absence of any liver insults but did not show any mitogenic response in IL-6 single transgenic mice. Thus, a signal induced via membrane-bound IL-6R on hepatocytes was insufficient to induce a proliferative response [108,109].

As shown in a model of acetaminophen-induced damage, MSCs significantly reduced necrosis through secreted growth factors in indirect co-cultures with the human LO2 cell line [110]. Many authors mistake the LO2 cell line for human hepatocytes, but it is a derivative of the cervical cancer cell line HeLa [111]. Several MSC-expressed factors, such as HGF, epidermal growth factor (EGF), IL-6, vascular endothelial growth factor (VEGF), and insulin-like growth factor-binding protein (IGFBP), may prevent hepatocyte apoptosis, increase hepatocyte survival, and/or stimulate proliferation [112,113]. In the presence of acetaminophen, the expression of some of these cytokines was markedly increased in human umbilical cord MSCs, and the addition of the cytokine-neutralizing antibodies to co-cultures led to a marked decrease in the anti-necrotic effect of the MSCs. It is noteworthy that among the analyzed cytokines, HGF made the greatest contribution to the protective effect of human umbilical cord MSCs on hepatocytes [110].

HGF is a pleiotropic factor that has mitogenic, motogenic, anti-apoptotic, morphogenetic, and immunoregulatory activities [114,115]. Such a wide range of activities warrants the ability of HGF to prevent fibrosis, inflammation, and apoptosis and promote tissue regeneration and angiogenesis [116]. It has been repeatedly shown that the therapeutic efficacy of MSCs in various pathologies partially or completely depends on the secretion of HGF [117,118]. The role of HGF in MSC-mediated therapeutic effects has been reliably proven in many studies using genetically modified MSCs with HGF overexpression [119]. Therapy with HGF-overexpressing MSCs has been shown to be more effective than using cells and cytokines separately or in combination [120,121]. HGF-overexpressing MSCs effectively reduced fibrosis and had a suppressive effect on hepatocyte apoptosis in liver disease models [121,122,123,124,125,126,127]. Also, HGF-overexpressing MSCs showed pronounced antioxidant activity, which is important for the alleviation of the inflammatory response through a reduction in oxidative stress by decreasing lipid peroxidation, probably via the ability to promote the activation of superoxide dismutase (SOD) and the synthesis of glutathione (GSH) [128].

Liu et al. [91] proved the role of MSC-produced prostaglandin E2 (PGE2) in reducing apoptosis and increasing hepatocyte proliferation caused by acute liver failure (ALF). The authors used LPS-treated AML12 cells as an in vitro model of ALF. MSC-CM and PGE2 significantly upregulated Ki67 and YAP expression and activated mTOR signaling in LPS-treated hepatocytes. Yes-associated protein (YAP), a transcriptional coactivator, is a core component of Hippo signaling that plays important roles in liver development, regeneration, and tumorigenesis (reviewed in [129,130,131]). When Hippo signaling is turned on, YAP is phosphorylated at serine S127 and then binds to the scaffolding molecule 14-3-3 and is transported to the proteasome for degradation. However, when Hippo signaling is turned off, YAP is translocated to the nucleus, where it binds the transcription factors of the TEA domain family (TEAD) to regulate genes associated with Hippo signaling [132,133]. In various models of liver injury, it has been shown that YAP activation improved organ regeneration by enhancing hepatocyte proliferation and reducing hepatocyte apoptosis [134,135]. In addition, the protective role of YAP has been shown in the regulation of hepatocyte ferroptosis in a sepsis-induced liver injury model [136]. The mammalian target of the rapamycin (mTOR) pathway regulates cell growth, survival, and metabolism [137]. mTOR plays a critical role in liver regeneration, hepatic lipid metabolism, and liver cancer [138,139]. Recent studies have shown a potential crosstalk between the Hippo and the mTOR pathways under normal physiological and pathological conditions [140]. Thus, the authors showed that MSC-produced PGE2 activates two interconnected signaling pathways involved in liver regeneration: YAP activation is mediated by p-CREB (p-cAMP responsive element binding protein); in turn, YAP activation leads to the suppression of PTEN due to miR-29a-3p and the subsequent activation of mTOR signaling, as evidenced by increased expression of p-S6K, p-AKT, and p-GSK3β [91]. Table 1 summarizes the data on the bioactive molecules and EVs produced by MSCs and protecting hepatocytes from apoptosis.

PGE2 is defined as a lipid-derived cytokine, an inflammatory lipid mediator that is formed from arachidonic acid by the action of cyclooxygenase 2 (COX2) [141]. PGE2 may act as a morphogen, regulating endodermal cell fate and specification in embryonic development, in particular determining the differentiation of endodermal progenitors into liver or pancreatic cells [142]. In the adult liver, PGE2 is produced by hepatocytes [143], Kupffer cells [144], and endothelial cells [145]. The role of PGE2 in the functioning of hepatocytes under normal and pathological conditions was demonstrated quite a long time ago. PGE2 has a significant effect on glycogenesis by stimulating the conversion of glucose to glycogen [146]. However, it also stimulates the breakdown of glycogen (glycogenolysis) in hepatocytes [147]. Specifically, PGE2 can either stimulate glycogen phosphorylase activity (glycogenolytic effect) or inhibit glucagon-stimulated glycogen-phosphorylase activity (antiglycogenolytic effect) in rat hepatocytes, depending on the type of receptor it binds to [148]. Prostaglandins can influence hepatocyte metabolism directly [149] or indirectly through hormones [150,151] or cytokines [152,153].

The effects of PGE2 on the hepatic lipid metabolism described in the literature are contradictory. Some studies have shown that prostaglandins can promote fat accumulation in hepatocytes, thereby facilitating the development of steatosis [154], while others have shown that PGE2 suppresses de novo lipogenesis [155] or does not affect lipogenesis but at the same time attenuates triglyceride incorporation into very-low-density lipoprotein (VLDL) [156]. Henkel et al. [157] have shown that PGE2 increases triglyceride accumulation in hepatocytes, most likely through the inhibition of β-oxidation combined with decreased VLDL production, thus contributing to hepatic steatosis. On the other hand, PGE2 has been shown to be involved in liver regeneration; its production levels in the liver increase biphasically during liver regeneration after partial hepatectomy, causing hepatocyte proliferation [158]. Rudnick et al. [159] showed that prostaglandins are essential for liver regeneration and COX-2, which is involved in the biosynthesis of PGE2, plays a particularly important role in this process. Hepatocyte-specific COX-2 transgenic mice showed reduced hepatocyte necrosis, decreased neutrophil infiltration, diminished oxidative and ER stress, and low levels of pro-inflammatory factors after liver ischemia/reperfusion injury (IRI) compared with wild-type mice [160]. A number of early studies have shown that PGE2 promotes the proliferation of primary rat hepatocytes [161,162] and has a mitogenic effect that is synergistic with EGF action [163,164]. Later, it was shown that hepatocyte proliferation occurs after the interaction of PGE2 with one of its EP3 receptors but not with the EP1 receptor [165]. However, in more recent studies, the stimulatory role of PGE2 in liver regeneration has been questioned. Nishizawa et al. [166] showed that the inhibition of inducible microsomal PGE synthase-1 (mPGES-1), the terminal enzyme of PGE2 generation, resulted in the stimulation of liver repair in animals subjected to hepatic IRI, whereas the activation of this enzyme exacerbated damage and delayed liver repair due to the accumulation of pro-inflammatory macrophages [166]. It has also been shown that COX2 is involved in the development of obesity [167] and fatty liver disease [168]. In the HBV-infected liver, endogenous PGE2 induced M1 polarization of macrophages and suppressed M2 polarization via EP4 receptors, leading to increased inflammation and aggravation of the nonalcoholic steatohepatitis (NASH) phenotype, while the EP4 antagonist, to some extent, improved the disturbance of lipid accumulation in the liver and balanced Kupffer cell polarization [169]. On the contrary, it was shown that in acute liver failure, the therapeutic potential of MSCs depended on the secretion of PGE2. MSC-derived PGE2 inhibited TGF-β-activated kinase 1 (TAK1) signaling and NLRP3 inflammasome activation in hepatic macrophages, resulting in decreased production of inflammatory cytokines. MSC-derived PGE2 via STAT6 and mTOR signaling induced the polarization of M2 macrophages, contributing to the resolution of inflammation and limiting liver damage. It is noteworthy that an EP4 antagonist reduced the M2 polarization of macrophages and their secretion of the anti-inflammatory cytokine IL-10 [170]. In their review, Cheng et al. [171] described in detail the mechanisms of PGE2-mediated tissue regeneration under various pathological conditions and PGE2-based therapeutic strategies. The authors explained the conflicting results regarding the role of PGE2 in IRI by the use of different model animals with varying pathological backgrounds, nonidentical ischemia reperfusion times, and diverse treatment methods, such as local or global I/R and warm or cold reperfusion.

#### 3.1.2. Involvement of miRNAs in the Anti-Apoptotic Effects of MSCs

MiRNAs play an important role in the normal development, function, and regeneration of the liver. MiRNAs are involved in the regulation of hepatocyte proliferation, differentiation, and metabolic activity. A review by Kweon et al. described in detail the spectrum of miRNAs that are produced in the liver and are critical for different liver functions [172].

Hydrogen peroxide (H_2_O_2_)-induced oxidative stress in LO2 cells resulted in an increased number of apoptotic cells and reduced mitochondrial membrane potential, as well as an increased percentage of the cell population in the G0/G1 phase and a sharp decrease in the S and G2/M phases, showing that H_2_O_2_ induces cell cycle arrest in LO2 cells. In addition, H_2_O_2_ significantly reduced the Bcl-2/Bax ratio and markedly increased BMF expression. The level of miR143 was significantly elevated in cells treated with H_2_O_2_. The targets of miR143 could presumably be ADRB1 (beta-1-adrenergic receptor) and HK2 (hexokinase 2), the expression of which was significantly reduced in H_2_O_2_-treated cells. MiR143 inhibited Bcl-2 expression and activated Bax and caspase-9, consequently activating the endogenous mitochondrial pathway that promotes cell apoptosis [173]. A study by Zhang et al. [174] showed that in osteosarcoma cells, Bcl-2 could be the target gene for miR143. In addition, in prostate cancer, miR143 was shown to target HK2 [175] and ADRB1 and play an important role in proliferation and apoptosis during H_2_O_2_ injury. It was shown that a decreased level of HK2 in hepatocellular carcinoma cells significantly inhibited glucose flux to pyruvate and lactate. Decreased glycolysis was associated with increased oxidative phosphorylation [176]. MiR143 can directly target the HK2 3’-UTR, thereby suppressing glucose uptake, lactate production, cellular G6P and ATP levels, and liver cell proliferation upon H_2_O_2_ injury. It was demonstrated that the hUC-MSC-CM reversed the negative effects of H_2_O_2_ on normal hepatocytes. In particular, hUC-MSC-CM reduced apoptosis in H_2_O_2_-treated LO2 cells by half, restored the mitochondrial membrane potential, increased the Bcl-2/Bax ratio, and reduced BMF expression within 3 h. Thus, hUC-MSC-CM had a pronounced anti-apoptotic effect under oxidative stress conditions, abolished cell cycle arrest, and promoted the transition of cells to the G1/S phase. In addition, hUC-MSC-CM reduced the miR143 level in cells subjected to oxidative stress and, accordingly, increased the levels of HK2 and ADRB1 [173]. The MSC-CM contains a large number of cytokines, promoting cell growth after H_2_O_2_ removal [177,178]. It can also be hypothesized that the hUC-MSC-CM affects H_2_O_2_-damaged LO2 cells by modulating glycolysis. In short, the promotion of LO2 cell proliferation and protection from apoptosis may occur through the regulation of the miR143/HK2 axis and the restoration of ADRB1 expression after hUC-MSC-CM treatment [173]. In the same model of oxidative stress, another miR was found, the level of which increased after LO2 cell treatment with H_2_O_2_, namely miR-486-5p [179]. The proviral integration site for Moloney murine leukemia virus kinase 1 (PIM1) was verified as the miR-486-5p target; accordingly, PIM1 expression significantly decreased with an increase in the level of miR-486-5p in LO2 cells subjected to oxidative stress; the TGF-β/Smad pathway was also activated in these cells. The HUC-MSC-CM attenuated damage caused by oxidative stress in LO2 cells by inhibiting miR-486-5p, upregulating PIM1, and blocking TGF-β/Smad signaling. Thus, the UCB-MSC-CM inhibits TGF-β/Smad signaling through the miR-486-5p/PIM1 axis [179].

In addition to reducing the expression of miRNAs involved in the progression of pathological states of hepatocytes, MSCs upregulate a number of miRNAs that have anti-apoptotic and pro-proliferative functions. However, the signals transmitted from MSCs to upregulate these miRNAs remain unclear. For example, Zhou et al. [180] showed that hUC-MSCs reduce apoptosis and enhance the proliferation of primary mouse hepatocytes isolated from bile duct ligation (BDL) livers by upregulating miR-148-5p, which plays an important role in hepatocyte differentiation and liver regeneration. It has been shown that miR-148 expression is downregulated in liver damage, and its upregulation leads to a pronounced therapeutic effect [181]. Human BM-MSC-derived extracellular vesicles (EVs) reduced hepatocyte apoptosis and promoted cell proliferation by upregulating miR-20a-5p, which targeted the the PTEN/AKT signaling pathway in mice with LPS/D-GalN-induced ALF [182].

The mechanisms of anti-apoptotic action of MSCs on damaged hepatocytes are shown in Figure 3.

In addition to regulating hepatocyte miRNAs, MSCs also produce various miRNAs that may mediate their regenerative effects on hepatocytes. The delivery of miRNAs from one cell to another is carried out by extracellular vesicles, in particular, by exosomes. Human chorionic plate-derived MSCs (CP-MSCs) were shown to suppress the development of CCl4-induced liver fibrosis in rats due to the increased content of miRNA-125b in exosomes, which inhibited Hedgehog (Hh) signaling in hepatic stellate cells, thereby slowing down fibrogenesis, and stimulated the accumulation of progenitor cells in the liver, inducing regeneration [183]. MiR-181-5p produced by adipose-derived MSCs was shown to exert the anti-fibrotic effect by suppressing the activation of hepatic stellate cells [184]. In a partial rat hepatectomy model, human umbilical cord blood mesenchymal stem cell (hUCB-MSC)-derived exosomal miR-124 promoted liver regeneration by enhancing hepatocyte proliferation through the downregulation of Foxg1 [185].

The detailed mechanisms of the anti-apoptotic effect of MSCs on hepatocytes remain unclear. However, it has been undeniably established that soluble hepatotrophic factors, such as HGF, IL-6, PGE2, etc., as well as extracellular vesicles loaded with protective bioactive molecules, can be considered as MSC apoptosis-reducing signals.

### 3.2. Regenerative Effects of MSCs

Liver regeneration is a well-studied process. The most common model for deciphering the mechanisms of liver regeneration is partial hepatectomy, which was extensively used to investigate the temporal and spatial processes involved in restoring organ mass, architecture, and function. In most cases, after partial hepatectomy, the liver is restored due to hepatocyte hyperplasia and hypertrophy. However, in real clinical situations, liver surgery does not lead to complete organ regeneration due to pre-existing pathological abnormalities; it entails a number of undesirable events leading to complications, including fibrosis, severe oxidative stress, etc. In this regard, MSCs may become valuable assistants in triggering full-fledged liver regeneration after surgical interventions, as demonstrated in many experimental studies involving the models of partial hepatectomy [70,186,187], liver transplantation [188,189], ischemia/reperfusion injury (IRI) [190,191], bile duct ligation (BDL) [192], etc. In all these studies, MSCs or MSC-derivatives have shown a pro-regenerative potential to reduce apoptosis of hepatocytes, increasing their proliferation and metabolic activity, and suppress inflammation in the liver by reducing infiltrating pro-inflammatory immune cells and inhibiting fibrosis by abolishing the activation of hepatic stellate cells. However, the majority of the MSC-derived factors associated with such a wide range of regeneration-related events remain unknown.

In a number of studies, it has also been demonstrated that MSCs or MSC-derivatives, inducing in liver recovery after surgery, do not always directly affect hepatocytes, or they did not have positive effects on the restoration of liver function at all. In a mouse model of partial hepatectomy, MSCs were shown to exert their therapeutic effect through the secretion of exosomes. However, it was shown that MSC-derived exosomes were not taken up by hepatocytes but were captured by macrophages, causing their polarization into M2 macrophages and thereby inhibiting inflammation [193]. The allogeneic MSC transplantation in a large animal model of repeated biliary obstruction followed by partial hepatectomy promoted the growth of liver tissue without any effect on liver function [194]. After partial hepatectomy of a fibrotic or cirrhotic liver, MSCs injected locally into the liver, but not those injected systemically, had a regenerative effect [195], although most studies have shown the therapeutic effects of MSCs and their derivatives, regardless of the transplantation route [196].

The secretome derived from the adipose tissue-derived MSCs pre-conditioned with the secretome taken from the damaged hepatocyte culture (MSC-iCM) showed higher therapeutic potential compared with the naïve secretome after injection into mice with the thioacetamide-induced hepatic failure and those with partial hepatectomy. Proteomic analysis showed that the MSC-iCM contained a large amount of antioxidant factors, in particular peroxiredoxin-1. After incubation with the MSC-iCM, damaged hepatocytes showed the highest expression of liver regeneration markers (HGF, VEGF, PCNA, and p-ERK) and the lowest expression of apoptotic and inflammation markers (PARP, BIM, BAX, and F4/80) [197]. Peroxiredoxin-1 catalyzes the reduction of H_2_O_2_ and alkyl hydroperoxide and thus protects cells from the attack of free radicals. Presumably, peroxiredoxin-1, produced by MSCs, interacts with TLR4 on the surface of hepatocytes, leading to an increase in the expression of the hepatoproliferative marker (p-ERK), thereby stimulating proliferation. Peroxiredoxin-1 levels also increased in damaged hepatocytes treated with the MSC-iCM [197]. The delivery of another antioxidant, glutathione peroxidase 3, by engineered human induced pluripotent stem cell-derived MSCs into the liver of mice with IRI and subsequent partial hepatectomy significantly suppressed hepatocyte senescence and apoptosis [198]. Together, these results demonstrate that MSCs can be a source of antioxidants to reduce oxidative stress in hepatocytes.

An interesting crosstalk mechanism between stressed hepatocytes and MSCs was revealed by Choi et al. using the bile duct ligation model, which is the most common surgical model used to induce cholestasis injury in rodents, causing proliferation of the biliary tract epithelial cells and oval cells and leading to inflammation, fibrosis, and hepatocyte apoptosis [199]. Human chorion plate-derived MSCs (CP-MSCs) have been shown to be effectively engrafted into an injured rat liver and facilitate the resolution of fibrosis, promoting liver regeneration after transplantation in rats undergoing bile duct ligation. In the in vitro experiments, the authors found reciprocal paracrine regulation of PRL-1 between rat hepatocytes treated with lithocholic acid (LCA), a bile acid used to mimic cholestasis, and CP-MSCs in an indirect co-culture system. The phosphatase of the regenerating liver (PRL) family is a group of protein tyrosine phosphatases (PTPs) that consists of three closely related members, PRL-1, PRL-2, and PRL-3, which have 76% to 87% amino acid identity [200]. PRL-1 is not a factor critical for liver development, postnatal growth, and hepatocyte differentiation; however, this tyrosine phosphatase is required to regulate the timing of liver regeneration after partial hepatectomy [201]. PRL-1 is an early growth response gene that is activated shortly after partial hepatectomy and triggers a cascade of regenerative events via the cytokine-independent pathway [202]. PRL-1 promotes G1/S progression by modulating the expression of cell cycle regulators via the activation of the AKT/STAT3 signaling pathway [203]. PRL-2 is expressed at high levels in several cell types, including hippocampal pyramidal neurons, ependymal cells, cone and rod photoreceptor cells, endocardium, vascular and bronchial smooth muscle, and collecting duct epithelium in the kidney. In parenchymal cells of the liver and pancreas, its expression is minute [203]. The role of PRL-2 in vivo [204] and in vitro [205] is modulation of the bioenergetic functions. Knockdown of PRL-2 leads to a significant decrease in intracellular levels of ATP and glutamine uptake. Subsequently, PRLs were also shown to have oncogenic roles. For example, overexpression of PRL-1 has been observed in several types of cancer and is correlated with poor prognosis, including gastric, intrahepatic, and prostate cancer [206,207,208]. PRL-3 is overexpressed in murine melanoma cells, where it plays a key role in cell motility and metastasis [209]. It also promotes endothelial cell migration in vitro [210]. PRL-3 is a mediator of metabolic reprogramming of cancer cells. PRL-3 increases glycolysis in various types of cancer, including myeloma [211,212] and colorectal cancer [213]. Luo et al. [214] showed that PRL-1 overexpression is associated with increased MMP-2 and MMP-9 levels in HEK293 cells through the Src and ERK1/2 pathways. CP-MSCs that expressed PRL-1 promoted liver regeneration through microRNA expression and mitochondrial dynamics in a liver cirrhosis rat model [215]. In the aforementioned work [199], the authors showed that normal hepatocytes and CP-MSCs both express PRL-1. After the treatment of primary rat hepatocytes with lithocholic acid, the expression level of PRL-1 decreased, while co-cultivation with CP-MSCs resulted in the restoration of its expression and subsequent regulation of MSC homing through RhoA-mediated ROCK1 signaling. PRL-1 inhibition by siRNA-PRL-1 reduced CP-MSC migration to damaged primary rat hepatocytes by reducing RhoA and ROCK1 levels as well as by decreasing MMP-2 levels. These results indicate that the expression of the Rho family and MMPs is regulated by PRL-1, which may be a key regulator for chemo-attractive CP-MSC migration (Figure 4). Genetically modified BM-MSCs overexpressing PRL-1 increased anaerobic mitochondrial metabolism in damaged hepatocytes, decreasing cytoplasmic lactate and increasing mitochondrial lactate, which ultimately led to increased ATP synthesis and hepatocyte repair [216]. The scheme shown in Figure 4 demonstrates the PRL-1-dependent mechanism of the MSC protective action.

Thus, hepatocyte proliferation, the process underlying liver regeneration, can be launched by the interaction of MSCs with damaged hepatocytes via the secretion of hepatotrophic factors, exosomes, and miRNAs. Reciprocal paracrine stimulation occurring between damaged hepatocytes and MSCs can lead to increased homing of the latter to the liver and the enhancement of their regenerative abilities.

### 3.3. Anti-Ferroptosis and Anti-Pyroptosis Effects of MSC

Ferroptosis is a type of “programmed necrosis” that was initially identified in cancer cells as nonapoptotic cell death mainly characterized by iron overload and lipid peroxidation [217]. Later, it was shown that cell death due to ferroptosis is characteristic not only to tumor cells, but it is also involved in the pathogenesis of certain diseases, such as neurodegeneration [218] and acute renal failure [219]. However, the role of ferroptosis in the development and progression of liver diseases remains poorly explored [220]. A number of studies have shown that ferroptosis may be a novel type of cell death associated with liver diseases, such as viral hepatitis, drug-induced liver injury [221], alcoholic liver disease [222], non-alcoholic fatty liver disease (NAFLD) [223], non-alcoholic steatohepatitis (NASH) [224], and hemochromatosis [225]. Tsurusaki et al. [224] identified the type of cell death that occurs at the earliest stage of NASH. NASH is a severe form of NAFLD characterized by the accumulation of lipid droplets in hepatocytes, hepatocyte death, inflammatory cell infiltration, and fibrosis. Earlier studies have shown that apoptosis and necroptosis are major contributors to the pathogenesis of NASH. However, the mechanisms underlying the transition from simple steatosis to steatohepatitis remained unclear until recently. In a mouse choline-deficient, ethionine-supplemented diet model of steatohepatitis, the authors proved that the onset of the disease was associated with massive necrosis of hepatocytes around the portal vein, which preceded apoptosis. They established that the necrotic death of hepatocytes is nothing more than ferroptosis and that ferroptosis is the initial cell death that induces inflammation at the onset of steatohepatitis [224]. From these results, it can be concluded that ferroptosis may be involved not only in the progression of liver diseases but also in their initiation.

As mentioned above, ferroptosis is a specific paradigm of cell death resulting from iron overload, reactive oxygen species generation, and lipid peroxidation, leading to antioxidative system dysfunction and, ultimately, cell membrane damage [226]. Ferroptosis is a form of regulated cell death driven by perturbation of the glutathione (GSH)-dependent lipid hydroperoxide scavenging network [227]. Lipid peroxidation is regulated by the X_C−_ system, which transports glutamate out of the cell in exchange for the transport of cystine, an important precursor for the synthesis of glutathione (GSH), into the cell [228]. The X_C−_ system consists of a light-chain subunit (SLC7A11/xCT) and a heavy-chain subunit (SLC3A2). SLC7A11 is a multi-pass transmembrane protein that operates as the cystine/glutamate antiporter. SLC7A11 transports cystine into the cell, where it is rapidly converted to cysteine, which is the rate-limiting precursor of GSH. Thus, SLC7A11 plays an important role in providing cysteine for glutathione biosynthesis. GSH is important for glutathione peroxidase 4 (GPX4), which protects cells from ferroptosis by converting toxic lipid hydroperoxides to non-toxic fatty alcohol [229].

Ferroptosis is a critical driver of CCl_4_-induced liver injury in mice, a commonly used in vivo model for acute liver injury [230]. MSCs, as well as MSC-derived exosomes, reduced ROS accumulation, downregulated malondialdehyde (MDA) levels, and increased SLC7A11 expression in damaged hepatocytes, thereby exerting a pronounced antioxidant effect, when indirectly co-cultured with primary mouse hepatocytes obtained from the livers of animals with CCl_4_-induced acute liver injury. At the same time, the MSC-Exo-induced recovery of SLC7A11 was accompanied by the increased expression of CD44 and OTUB1 on hepatocytes. OTUB1 is an ovarian tumor deubiquitinase (OTUB) family member that regulates the SLC7A11 stability through direct protein interaction [231]. CCl_4_ increased SLC7A11 ubiquitinylation, thereby abolishing its stabilization and leading to its downregulation, while MSC-Exo downregulated SLC7A11 ubiquitinylation, restoring its stabilization, which led to the upregulation of SLC7A11. Additionally, SLC7A11 was stabilized by MSC-Exo-mediated upregulation of CD44 and OTUB1 expression, both of which coprecipitate with SLC7A11 [230]. Several studies have shown that CD44 stabilizes SLC7A11 expression in the plasma membrane, thus promoting cystine uptake and GSH synthesis [228].

In a rat fatty liver transplantation model, exosomes derived from heme oxygenase 1-overexpressed human BM-MSCs were shown to abolish hepatocyte ferroptosis, improve liver function, inhibit Kupffer cell and T cell activation, and prevent long-term biliary fibrosis [232]. Exosome transplantation reduced Fe^2+^ levels and molecular markers of ferroptosis, such as Ptgs2 mRNA, 4-hydroxynonenal (4-HNE), and malondialdehyde (MDA), which were significantly elevated in animals without exosome treatment. The level of glutathione, an important antioxidant maintaining cellular redox homeostasis, also increased strongly after exosome transplantation. The key negative regulator of ferroptosis in this model was miR-204-5p, which is highly expressed in exosomes derived from heme oxygenase 1-modified human BM-MSCs. The miR-204-5p target is ACSL4, a key regulator of ferroptosis, whose expression was strongly reduced after treatment with exosomes. Simultaneously, the expression of GPX4, a critical protective enzyme that prevents ferroptosis, was upregulated [232]. That is, exosomes obtained from heme oxygenase 1-modified human BM-MSCs, due to miR-204-5p, not only inhibit ferroptosis but also activate the antioxidant system. In the I/R myocardial injury model, it was shown that exosomes derived from hUCB-MSCs and loaded with exosomal miR-23a-3p inhibited ferroptosis by regulating iron metabolism independent of the GPX4 pathway [233]. The role of exosomes in ferroptosis-related pathogenic mechanisms in various diseases is described in detail in this review [234]. Figure 5 shows the scheme of hepatocyte protection from ferroptosis due to MSC and MSC derivatives.

Pyroptosis, a rather recently characterized type of programmed cell death, plays an important role in various liver diseases. Pyroptosis is characterized by caspase-dependent (caspases 1, 4, 5, and 11) pore formation in the cell plasma membrane and the subsequent release of inflammatory mediators (interleukin-18, IL-1β). Inflammasomes are caspase activators in pyroptosis. Inflammasomes are multiprotein complexes that activate caspase 1 in response to stress signals. Cleaved caspase 1 mediates the maturation and secretion of IL-1β and IL-18 and pyroptotic cell death. The hallmarks of pyroptosis are plasma membrane hyperpermeabilization, cell swelling, rapid cell lysis, and the release of cytoplasmic contents and pro-inflammatory cytokines into the microenvironment. Initially, pyroptosis was considered an innate immune mechanism against intracellular bacteria. However, there is increasing evidence that it is also involved in sterile inflammation, such as that in acute or chronic liver diseases. More information about the mechanisms of pyroptosis and its role in liver diseases has been described in several recent review articles [235,236]. Inflammasomes are predominantly expressed by immune cells, although many studies have shown that hepatocytes undergoing pyroptosis due to liver damage release inflammasomes, which are taken up by other cells and thereby mediate inflammatory and pro-fibrogenic stress signals [237].

Recent studies have shown that inflammasome-mediated pyroptosis is a driver of inflammation and plays a critical role in the progression of liver pathologies, such as autoimmune liver diseases [238], liver fibrosis [239], and fatty liver diseases [240,241]. Therefore, pharmacological pyroptosis targeting may become an option for reducing inflammation in liver diseases. It has been found that MSCs or MSC derivatives are able to suppress inflammation by inhibiting pyroptosis [242,243]. In a mouse model of acute liver failure induced by D-galactosamine, the injection of mouse BM-MSCs significantly reduced the expression of the NLRP3 inflammasome protein and the production of pro-inflammatory cytokines associated with the progression of pyroptosis, IL-1β, and IL-18. Similar results were obtained with D-galactosamine-treated primary mouse hepatocytes after indirect co-culturing with mouse BM-MSCs, in which MSCs markedly reduced NLRP3-caspase 1 inflammasomes in hepatocytes. The main driver of the anti-pyroptosis effect of MSCs was shown to be the anti-inflammatory cytokine IL-10, which inhibits the expression of NLRP3, thereby negatively regulating inflammasome-associated pyroptosis and inflammation [244]. In a mouse model of CCl_4_-induced liver cirrhosis, a triggering role of pyroptosis in the progression of liver injury was also shown, as demonstrated by the high expression of gasdermin D (GSDMD), caspase 1, and NLRP3 and the pro-inflammatory cytokine IL-1β in the animals’ livers. GSDMD, the key pyroptosis executioner, was shown to be the only pyroptosis effector with pore-forming activity [245]. Stem cells isolated from human exfoliated deciduous teeth attenuated liver cirrhosis in animal models by inhibiting pyroptosis. In vitro experiments using the co-cultivation of mouse hepatocytes (mouse hepatic cell line NCTC 1469) treated with CCl_4_ with human exfoliated deciduous teeth-derived stem cells in the transwell system showed that the pyroptosis-inhibiting ability is associated with decreasing ROS levels and normalization of the membrane mitochondrial potential in hepatocytes induced by soluble factors produced by MSCs [246]. In a mouse model of acute liver failure induced by D-galactosamine (D-GalN) and lipopolysaccharide (LPS), transplantation of human UC-MSCs was shown to inhibit not only apoptosis and inflammation but also pyroptosis by suppressing NLRP3 inflammasome activation, IL-1β maturation, and caspase 1 cleavage [88]. Palmitic acid-treated HepG2 (an in vitro model of fatty liver disease) significantly increased the expression of the characteristic markers of pyroptosis, including NLRP3, GSDMD/-N (activated form of GSDMD), caspase 1, p20 (activated form of caspase 1), IL-1β, and IL-18, while indirect co-cultivation with MSCs significantly reduced their expression and suppressed pyroptosis [247]. Figure 6 shows the scheme of hepatocyte protection from pyroptosis by MSC and MSC derivatives.

Thus, both in liver injuries in vivo and in the course of indirect co-cultivation with hepatocytes subjected to pyroptosis-inducing stress in vitro, MSCs are able to protect hepatocytes from pyroptosis, most likely due to the secreted factors. For example, human UC-MSCs abrogated traumatic brain injury-induced pyroptosis in vivo and lipopolysaccharide/ATP-induced BV2 microglial pyroptosis in vitro due to the secretion of anti-inflammatory factors, such as tumor necrosis factor-stimulated gene 6 (TSG-6) [248]. The inhibition of pyroptosis in dextran sulfate sodium-induced ulcerative colitis in mice by transplantation of hair follicle-derived MSCs is critically associated with exosomes produced by mesenchymal cells since the inhibition of exosome secretion completely abolishes the protective effects of the MSCs in vivo and in vitro. Presumably, MSC-derived exosomes might inhibit pyroptosis through tumor necrosis factor-related apoptosis-inducing ligand (TRAIL) signaling and the IFN-γ pathway [249]. Also, many studies have shown that miRNAs transported by MSC-derived exosomes play a role in the suppression of pyroptosis caused by inflammation or other stimuli [250].

### 3.4. MSCs Protection from Lypotoxicity and ER Stress

Hepatocytes are the main cells in the body that carry out lipid metabolism. Normal lipid metabolism in hepatocytes includes their synthesis, uptake, efflux, and oxidation. These highly regulated processes maintain homeostatic intracellular lipid levels. Fatty acid transport into hepatocytes requires the action of the fatty acid-binding protein (FABPpm) and fatty acid translocase (FAT)/CD36. Some of the incorporated fatty acids undergo β-oxidation in the mitochondria by carnitine palmitoyltransferase 1a (CPT1A) [251] and mitochondrial trifunctional protein, which catalyzes the last three steps of the mitochondrial beta-oxidation of long-chain fatty acids, hydroxyacyl-CoA dehydrogenase/3-ketoacyl-CoA thiolase/enoyl-CoA hydratase alpha/beta subunit (HadhA/B) [252], followed by ATP generation. Another part of incorporated fatty acids is stored in hepatocytes in the form of fat droplets formed due to the action of endoplasmic reticulum-bound stearoyl-CoA desaturase 1 (SCD1), the key determinant of triglycerides biosynthesis pathway [253]; acetyl-CoA carboxylase 1 and 2 (ACC1 and ACC2), which regulate hepatic fat storage [254]; and fatty acid synthase (FAS), which catalyzes the de novo synthesis of fatty acids producing fat for the storage of energy when nutrients are present in excess [255].

Lipotoxicity associated with abnormal accumulation of lipids in hepatocytes underlies the pathogenesis of liver diseases such as non-alcoholic fatty liver disease (NAFLD) and non-alcoholic steatohepatitis (NASH) [256]. It was also shown that within a short time after liver surgery, the aberrant metabolism and abnormal accumulation of lipids are triggered in hepatocytes, eventually leading to liver injury and liver failure [257].

The abnormal accumulation of lipids in hepatocytes can occur due to stressful conditions, such as overexposure to saturated fatty acids or insulin resistance, which leads to the triggering of several pathological signaling mechanisms, including the accumulation of ROS, ER stress, and chronic inflammation. The endoplasmic reticulum (ER) performs important functions related to the synthesis, folding, and transport of proteins and plays a critical role in lipid synthesis and Ca^2+^ homeostasis [258]. Prolonged excessive uptake of saturated fatty acids leads to an imbalance between ER protein loading and folding, leading to the hyperactivation of the unfolded protein response (UPR) and the subsequent activation of the cell death pathway [259]. Disruption of ER calcium balance is one of the initial and key events of cell death induced by ER stress [260]. Impaired cellular Ca^2+^ homeostasis is a hallmark of many ER-related diseases and a key trigger for NAFLD/NASH. The sarcoplasmic/endoplasmic reticulum Ca^2+^ ATPase (SERCA) transports Ca^2+^ from the cytosol to the ER lumen, maintaining resting calcium concentrations. Saturated fatty acids inhibit the activity of SERCAs, which leads to the release of calcium from the ER lumen and a high calcium load in the cytosol, ultimately leading to the disruption of ER homeostasis and cellular function [261].

Several studies have shown that MSCs attenuate lipotoxicity in hepatocytes, which is associated with aberrant lipid accumulation and lipid metabolism disorders. However, the mechanisms of the attenuation of lipotoxicity in hepatocytes induced by MSCs remain poorly understood. Using a mouse natural aging model, Ling et al. [262] demonstrated that hUCMSC-exos significantly reduced aging markers and genomic instability in aging livers. Metabolome studies found that hUCMSC-exos reduced lipids associated with lipotoxicity and inflammation, including saturated glycerophospholipids, palmitoyl-glycerols, and eicosanoid derivatives. Using phosphoproteomics approaches, the authors also showed a decrease in the phosphorylation of the metabolic enzyme propionate-CoA ligase. In addition, it was shown that hUCMSC-exos significantly reduced the phosphorylation of proteins involved in nuclear transport and cancer signaling in hepatocytes, including heat shock protein HSP90-beta (Hsp90ab1) and nucleoprotein TPR, and increased the phosphorylation of proteins involved in the intercellular communication, such as calnexin and PDZ domain-containing protein 8 (Pdzd8) [262].

In the HepG2 cell line, palmitic acid (PA) was shown to reduce viability and increase apoptosis and lipotoxicity [247]. PA, a long-chain saturated fatty acid, undergoes β-oxidation in the mitochondria, and PA-induced lipotoxicity can cause mitochondrial dysfunction, which indicates bioenergetics disruption [263]. PA-induced lipotoxicity in HepG2 was manifested by the accumulation of massive fat droplets and an increase in the triglyceride level. The amount of the sterol regulatory element binding protein 1c (SREBP1c) (truncated SREBP1), which is a master transcription factor controlling lipogenesis and lipid uptake, was significantly increased in PA-treated hepatocytes. The indirect co-cultivation of PA-treated HepG2 with MSCs markedly inhibited the accumulation of fat droplets, reduced triglyceride levels, and decreased SREBP1c expression in hepatocytes. Thus, MSCs protected HepG2 from PA-induced lipid accumulation [247]. PA-induced lipotoxicity in hepatocytes resulted in the upregulation of a gene associated with fatty acid transport, CD36l the lipid synthesis genes, including PPARalpha, PPARgamma, FASN, and SREBP1c; as well as CPT1A, a mitochondrial enzyme responsible for the transport of fatty acids into the mitochondria for β-oxidation, while MSCs reduced their expression. Under lipotoxic conditions, abnormal morphological changes in the ER occurred in HepG2, including cistern enlargement and ribosome loss, which were not manifested during co-cultivation with MSCs. Compared with PA-treated monocultures, the co-cultures of HepG2 with MSCs showed reduced expression of typical ER stress marker proteins, including BiP, ATF6, ATF4, p-eIF2a, and CHOP, and genes associated with the UPR pathway, such as GRP78, FKBP11, and GRP94. MSCs restored the calcium balance between the ER and the cytosol, which was disturbed by the treatment of hepatocytes with palmitic acid by increasing the expression and activation of SERCA [247].

Sirtuin 1 (SIRT1) is an NAD+-dependent deacetylase that is a master regulator of the transcriptional network that controls hepatic lipid metabolism. Under energy depletion conditions, SIRT1 deacetylates and alters the expression of key transcriptional regulators involved in hepatic lipogenesis, fatty acid β-oxidation, and cholesterol/bile acid metabolism [264]. Human UC-MSC-CM improved mitochondrial function and reduced inflammation and apoptosis in the PA-induced steatosis in LO2 cells by reducing the levels of the pro-inflammatory cytokines TNFα, IL1β, and IL6 and increasing the protein levels of SIRT1, PGC1α, NRF1, and TFAM, the key molecules in maintaining mitochondrial function. Since after SIRT1 silencing in hepatocytes, the effect of the UC-MSC-CM on the levels of SIRT1, PGC1α, NRF1, and TFAM was significantly reduced and ROS production and the levels of pro-inflammatory cytokines increased, the authors suggested that SIRT1 mediates the protective effects of the UC-MSC-CM [265]. SIRT1 regulates cellular metabolism and mediates the deacylation of target proteins, such as peroxisome proliferator-activated receptor gamma coactivator 1α (PGC1α), which is involved in fatty acid oxidation and mitochondrial function [266]. SIRT1 overexpression effectively reduces obesity and insulin resistance in NAFLD rodents [267]. In patients with obesity or NAFLD, SIRT1 levels are significantly reduced in the plasma and in the liver [268]. Thus, SIRT1 and SERCA can serve as targets in the treatment of fatty liver diseases.

PA-treated mouse hepatocytes accumulated lipid deposits, in particular triglycerides and cholesterol, similar to the situation seen in NAFLD. After treating the hepatocytes with murine adipose-derived MSC (AMSC)-EVs, marked reduction of lipid deposits, normalization of lipid metabolism, and a reduction in liver fibrosis marker proteins (α-SMA, COL1A1, and TGF-β1) was observed. AMSC-EVs efficiently accumulated in hepatocytes over time and not only delivered miR-223-3p to them but also induced the upregulation of this miRNA in the recipient hepatocytes [269]. There were reports showing the involvement of granulocyte-enriched miR-223 [270] in chronic liver injury, such as non-alcoholic steatohepatitis (NASH) and acute hepatitis [271]. An miR-223 analog, miR-223-3p, improves acute and chronic hepatitis through inhibition of NLRP3 inflammasome activation [272]. BM-MSC-derived extracellular vesicles carrying miR-223-3p alleviate autoimmune liver disease by reducing the inflammatory response [273]. miR-223-3p has also been shown to have a mitigating effect against inflammatory diseases by targeting the inhibition of the E2F transcription factor 1 (E2F1) [274]. In turn, E2F1 modulates cholestatic liver fibrosis, highlighting its promise as a fibrogenic marker [275].

In a rat model, 90% hepatectomy induced acute steatosis, which was attenuated after the transplantation of rat BM-MSCs [276]. The expression analysis of the genes associated with different aspects of lipid metabolism, including lipid decomposition (CPT1a, Hadha, and Hadhb), lipid synthesis (Scd1, Acaca, and FASN), and lipid transport (CD36), showed that BM-MSCs mainly influenced the processes of lipid decomposition and metabolism but did not influence lipid synthesis and transport. The expression of lipolysis-related proteins, including peroxisome proliferator-activated receptor alpha (PPARα), CPT1a, and carnitine octanoyltransferase (CROT), increased after MSC transplantation, although there were no significant changes in the expression of lipid synthesis-related proteins, such as ACC and FASN. The expression of the sterol-regulatory element-binding proteins (SREBP1 and SREBP2) was also restored, although the expression of the FASN downstream protein did not change significantly. These results can be explained by the fact that BM-MSCs reduced lipid accumulation and restored lipid metabolism. PPARα, the key transcription factor regulating fatty acid β-oxidation, was upregulated after BM-MSC transplantation, showing that BM-MSC transplantation can improve fatty acid β-oxidation function. BM-MSC transplantation activated the mTOR pathway, and its activation played a critical role in liver recovery after failure caused by partial hepatectomy. The mTOR pathway may also be involved in the effects of MSCs on mitochondria and lipid metabolism. IL-10 was identified as the key BM-MSC-produced factor determining their protective action on hepatocytes subjected to lipotoxicity caused by 90% hepatectomy, as the depletion of this cytokine in BM-MSCs abolished the therapeutic effect [276]. IL-10 is a cytokine that regulates the growth and differentiation of innate immune cells. IL-10 can control cell proliferation, apoptosis, angiogenesis, and inflammation [277,278]. IL-10 is associated with the occurrence of lipid metabolism disorders [279]. IL-10 provides a key link between inflammatory factors and lipid metabolism. IL-10 mainly affects genes associated with lipid catabolism (CPT1a, Hadha, and Hadhb) and proteins associated with lipid β-oxidation (PPARα, CPT1a, and CROT). These results suggest that MSC-derived IL-10 is an important cytokine improving mitochondrial damage in partial hepatectomy liver failure. Also, IL-10 depletion inhibits mTOR pathway activation after MSC transplantation. Thus, MSCs can improve lipid metabolism, mainly by secreting IL-10 and influencing mitochondrial function [276].

MSCs ameliorate lipotoxic kidney injury via a novel microenvironment-dependent paracrine HGF/c-Met signaling mechanism to suppress ER stress and its downstream pro-inflammatory and pro-apoptotic consequences [280]. MSCs were able to attenuate endothelial lipotoxicity by inhibiting ER stress and the endothelial-to-mesenchymal transition by the secretion of stanniocalcin-1 (STC-1) [281].

The above-described data suggest that MSCs can offset lipotoxicity by secreting soluble factors, such as IL-10 or miR-223-3p, which inhibit ER stress and normalize mitochondrial function with the subsequent restoration of lipid metabolism.

There are various ways to increase the therapeutic activity of MSCs, including hypoxic conditions, 3D culturing, pre-treatment of mesenchymal cells with pro-inflammatory cytokines, lipopolysaccharides, and chemical agents [282]. All these methods lead to changes in the MSC secretome, promoting more pronounced immunomodulatory, regenerative, and proangiogenic properties. For example, hypoxic conditions cause the upregulation of prostaglandin E synthase (PTGES) in cultured MSCs, resulting in enhanced PGE2 production, which promotes the switch of macrophages to the anti-inflammatory M2 phenotype. Also, hypoxia upregulates miR120 in MSCs, which reduces hepatocyte apoptosis [283]. In mice, after radical hepatectomy, the hypoxia-induced upregulation of VEGF in transplanted MSCs promotes hepatocyte proliferation and liver regeneration [284]. The conditioned media obtained from the cultures of the adipose-derived stem cells pretreated with LPS contained higher levels of cytokines involved in liver regeneration, including interleukin-6 (IL-6), tumor necrosis factor-alpha (TNF-α), hepatocyte growth factor, and vascular endothelial growth factor. These conditioned media had a more pronounced supportive effect on mouse hepatocytes treated with thioacetamide, increasing their viability compared with the conditioned media obtained from the cultures of MSCs not treated with LPS. In vivo, in partially hepatectomized mice, the LPS-CM significantly enhanced liver regeneration, as manifested by a decrease in serum levels of aspartate transaminase and alanine transaminase [285]. Icariin-treated hUMSCs increased the antioxidant activities in the liver and prevented fibrosis progression in CCl_4_-induced chronic liver injury in mice [286].

### 3.5. Mitochondrial Transfer and Antioxidant Effects of MSCs

The main function of the mitochondria is to produce energy in the form of ATP. Furthermore, mitochondria are producers of superoxide anion radical (O2^•−^), from which reactive oxygen species (ROS) are formed. ROS are essential for regulating cell signaling and maintaining redox homeostasis [287]. Redundant superoxide anion radical (O2^•−^) and ROS can cause oxidative stress. Mitochondria contain antioxidant systems to quench oxidative stress. For example, manganese superoxide dismutase (MnSOD) converts O2^•−^ to H_2_O_2_ and molecular oxygen. Mitochondrial antioxidants, including GSH and thioredoxin, are able to buffer H_2_O_2_. The fission and fusion of mitochondria are required for the cell cycle and apoptosis progression as well as the control of their own functional integrity [288]. Together with the ER, mitochondria are involved in the storage and buffering of calcium. An elevated mitochondrial Ca^2+^ level may trigger cell death, including apoptosis, necrosis, and autophagy [289]. As described above, both the ER and mitochondria regulate lipid metabolism. Mitochondria serve as a site for the β-oxidation of fatty acids, resulting in the generation of acetyl-CoA [290]. In hepatocytes, mitochondria perform unique functions, including the regulation of gluconeogenesis, ammonia detoxification, and anabolic pathways [291]. Therefore, the disruption of the mitochondria functions in hepatocytes leads to general damage to the whole organism’s homeostasis. The most common types of mitochondrial dysfunction in liver diseases include the enhancement of the oxidative processes and/or impairment of the antioxidant mechanisms, leading to oxidative stress [292]. Mitochondrial dysfunction is observed in almost all liver pathologies, including alcoholic liver disease [293], NAFLD [294], viral hepatitis [295], fibrosis, and cirrhosis [296].

In various disease models, it has been shown that MSCs are able to restore the oxidant/antioxidant balance in damaged tissues and cells. Intravenously transplanted human UC-MSCs normalized mitochondrial function in a rat model of D-galactose-induced liver injury. This animal model is characterized by a high level of oxidative stress and mitochondrial dysfunction, which is manifested by a decrease in antioxidant capacity as well as the disturbance of bioenergetic functions, as evidenced by the loss of the mitochondrial membrane potential, increased ROS production, a decreased number of mitochondrial respiratory complexes, and decreased ATP. Human UC-MSCs restored the mitochondrial antioxidant system mainly by increasing the expression of SOD, GPx, and GSH via the activation of the nuclear factor erythroid 2-related factor 2/heme oxygenase-1 (Nrf2/HO-1) pathway [297]. In a mouse model of CCl_4_-induced liver damage, syngeneic BM-MSCs also had a pronounced antioxidant effect and restored mitochondrial function by increasing the gene expression levels of Hmox-1 (heme oxygenase-1), BI-1 (Bax inhibitor-1), HGF (hepatocyte growth factor), GST (glutathione transferase), and Nrf2 [298]. In the same model of CCl_4_-induced liver injury, human UC-MSC-Exo exerted antioxidant and anti-apoptotic effects [299]. In the above-described works, the mechanisms of the antioxidant effects of MSCs were not established, and the key molecular factors that were produced by MSCs and were capable of implementing their protective capabilities were not elucidated. That is to say, the question of the communication mechanisms between MSCs and damaged hepatocytes and their dysfunctional mitochondria remains open.

Some studies have suggested that such mechanisms can be mediated by MSC-produced extracellular vesicles. In the liver IRI model, human UC-MSC-derived extracellular vesicles protected against hepatic apoptosis by reducing neutrophil infiltration and alleviating oxidative stress in the liver tissue [300]. In the in vitro experiments, the authors found that human UC-MSC-EVs containing high levels of the MnSOD could transport this antioxidant enzyme into H_2_O_2_-damaged cells (LO2 cell lines), thereby increasing its levels in recipient cells and, as a result, restoring the oxidant/antioxidant balance. Importantly, siMnSOD-EVs derived from MSCs with quenched MnSOD expression did not have antioxidant or antiapoptotic effects in vitro or in vivo. Thus, MnSOD is the key protein mediating the protective functions of human UC-MSC-EVs by reducing oxidative stress and the apoptosis of hepatocytes in the IRI model [300]. Another antioxidant protein, glutathione peroxidase 1 (GPx1), has been shown to be delivered by MSC-derived exosomes into damaged hepatocytes, restoring mitochondrial function and reducing oxidative stress [301]. In the mouse model of CCl_4_-induced liver failure, human UC-MSC-derived exosomes reduced pro-inflammatory cytokines (G-CSF, IL-1α, IL-6, monocyte chemoattractant protein-1 (MCP-1), and TNF-α), and inhibited oxidative stress and apoptosis. In a model of H_2_O_2_ and CCl_4_-induced oxidative stress in LO2 cells, it was shown that human UC-MSC-derived exosomes abolished oxidative stress-induced apoptosis by inducing anti-apoptotic events, ERK1/2 phosphorylation, and Bcl2 expression and inhibiting the pro-apoptotic IKKB/NFkB/casp9/3 pathway. Eventually, it was proven that GPx1 antioxidant enzyme delivery into damaged hepatocytes or liver tissue by human UC-MSC-derived exosomes is the prime process underlying the reduction of oxidative stress and, as a result, leading to the prevention of hepatocyte apoptosis in vitro and in vivo [301].

A reduction in oxidative stress and restoration of mitochondrial function in liver diseases can also be achieved through the direct transfer of MSC-derived mitochondria to damaged hepatocytes. Many studies have demonstrated that MSCs can renew mitochondria in damaged cells through the transfer of their own mitochondria into them [302,303,304]. Transferred mitochondria are able to support the affected cells by improving mitochondrial quality control, the maintenance of macromolecular biosynthesis, and the cytoplasmic pool of NAD+ [304,305]. It was shown that the transfer of mitochondria between MSCs and damaged tissues is crucial for tissue regeneration [306]. Several studies have demonstrated that the mechanism of mitochondrial transfer from MSCs to damaged hepatocytes is a key event in the communication between the two cell types, leading to the restoration of damaged liver tissue. For example, in oxygen–glucose deprivation conditions, the human LO2 cell line underwent apoptosis and accumulated high levels of ROS. Co-cultivation of the LO2 cells with human UC-MSCs in various ratios abolished apoptosis and reduced the ROS level due to the transfer of mitochondria from MSCs into LO2 cells [307].

In NASH livers, lipid metabolism switches from utilization to storage. A hallmark of NASH is the elevation of liver lipid deposition due to excess fatty acid supply from adipose tissue coupled with impaired mitochondrial β-oxidation in the hepatocytes, involving reduced respiratory chain activity and impaired triglyceride secretion via low-density lipoprotein (VLDL) [308]. MSCs support the restoration of tissue homeostasis by attenuating NASH-associated pathomechanisms, such as lipid loading, oxidative stress, acute damage response, and the epithelial-to-mesenchymal transition. Since mitochondrial deterioration may be the cause of lipid accumulation, it is hypothesized that MSCs may improve mitochondrial function. Hsu et al. [75] revealed the lipolytic effect of human BM-MSCs pre-differentiated into a hepatocyte phenotype in the methionine–choline-deficient diet-induced NASH mouse model. The authors suggested that this protective effect may be related to the restoration of mitochondrial function in hepatocytes. The authors created an in vitro model of fatty hepatocytes by culturing primary mouse hepatocytes in a steatosis-inducing medium containing a methionine–choline-deficient medium or palmitic acid (C16:0). Under such conditions, hepatocytes accumulated large numbers of fat droplets. A decrease in hepatocyte lipid content was observed when fatty hepatocytes were directly co-cultured with human MSCs, indicating that MSCs really correct lipid metabolism. The lipolytic effect of MSCs was not manifested when the conditioned medium was added to hepatocytes, which indicates that intercellular contact, rather than soluble factors or exosomes, is necessary for the effect to be realized. The authors found that MSCs communicated directly with hepatocytes in co-cultures through long filopodia-like tubules derived from MSCs and contact with fatty hepatocytes or other MSCs. These tubules were enriched in F-actin, which is one of the essential features of tunneling nanotubes (TNTs). TNTs carry out the exchange of molecular and corpuscular messages between cells, and mitochondria are the usual cargo of TNTs [309,310]. Notably, the cargo exchange between fatty hepatocytes and MSCs is bidirectional. However, the rate of cargo transportation from hepatocytes to MSCs proved to be somewhat lower (627 nm/min) than from MSCs to hepatocytes (1656 nm/min). In addition to mitochondria, peroxisomes are also delivered from MSCs to hepatocytes via TNTs. Human BM-MSCs were found to contain mouse mitochondria after co-culture with mouse fatty hepatocytes. It is not certain that the transfer of mouse mitochondria was also carried out via TNTs. However, the reciprocal exchange of organelles between different types of cells even in xenogeneic systems is a proven fact. An important therapeutic result of the transfer of MSC-derived mitochondria and peroxisomes to fatty hepatocytes was the restoration of lipid metabolism through the two following mechanisms: (1) the activation of the key regulators of lipid utilization, including peroxisome proliferator-activated receptor α (PPARA), by stimulating mitochondrial biogenesis in hepatocytes, as shown by increased co-culture expression of mouse PGC1α (PPARGC1A; PPARγ coactivator 1 α), a regulator of mitochondrial biogenesis; (2) the increase in the lipid-oxidizing capacity in hepatocytes due to the increased numbers of MSC-derived mitochondria in the hepatocytes. Taken together, these results show that human MSCs previously differentiated into the hepatocyte phenotype can deliver mitochondria (and bona fide peroxisomes) to mouse hepatocytes, which in turn can support lipid breakdown both by providing oxidative capacity and by supporting the ability of hepatocytes to utilize lipids, potentially supporting mitochondrial biogenesis. Human MSCs transplanted into mice intrahepatically were shown to donate their mitochondria to host hepatocytes [75]. Using another high-fat-diet-induced NASH model, the same authors later found that mitochondrial donation from human MSCs to hepatocytes represents the primary level of MSC action that secondarily mediates universal mechanisms involved in improving NASH, regardless of disease etiology [311]. Figure 7 illustrates the mitochondrial transfer from MSCs to hepatocytes and explains the recovery mechanisms associated with this process.

In a mouse T2DM-associated NAFLD model, BM-MSC transplantation reversed the abnormally elevated alanine aminotransferase (ALT), aspartate aminotransferase (AST), triglycerides (TGs), and total cholesterol levels; completely reversed steatosis; and restored to normalcy, such hepatocyte mitochondria characteristic morphology, mtDNA copy number, oxygen consumption rate, and ATP production. Also, BM-MSC treatment restored malondialdehyde (MDA), ROS, and mitochondrial membrane potential. The authors showed that in the hepatocytes of the recipient T2DM mice, mitochondria from transplanted MSCs were localized near the nuclei in the cytoplasm of hepatocytes. The mitochondrial transfer from MSCs was confirmed in vitro using HepG2 cultured in the free fatty acid medium. Notably, the negative consequences of T2DM-associated NAFLD, namely steatosis and associated pathophysiological processes, including lipid accumulation, oxidative stress, mitochondrial dysfunction, and decreased ATP levels, were not reversed when mitochondrial transfer from BM-MSCs was blocked [312]. These results suggest that mitochondrial transport, rather than paracrine signaling, plays a central role in the rescue of the steatotic hepatocytes. Another study showed that mitochondrial transfer is the initial mechanism that triggers the processes of restoration of the redox homeostasis of the cell, the normalization of lipid metabolism, and, as a result, the prevention of hepatocyte apoptosis and the subsequent development of inflammation in the damaged liver.

It is noteworthy that at present, the direct targeted delivery of mitochondria to damaged hepatocytes is increasingly being considered as a possible therapeutic approach for the treatment of liver diseases (see more in the review [313]).

## 4. Conclusions

The data reviewed in this article clearly show that the interaction and mutual influence of hepatocytes, liver parenchymal cells, and MSCs or MSC-like stromal cells present in the liver are crucial for liver tissue integrity and function in health and disease. The interaction involves direct cell-to-cell contact and paracrine mechanisms using soluble factors and extracellular vesicles as messages. Direct contact takes place not only between closely adjacent cells but also via long processes formed by MSCs. A substantial amount of the information concerning the MSC-hepatocyte crosstalk comes from in vitro experiments. The in vivo verification of these data remains partly inconsistent due to the interference of many cell types, including hepatic stellate cells, macrophages, and infiltrating immune cells, as well as other biologically active microenvironmental components, such as extracellular matrix proteins and blood-born factors.

The analysis of the hepatocyte–MSC interaction shows that MSCs carry out chaperone-like functions, including the synthesis of the supportive extracellular matrix proteins, prevention of apoptosis, pyroptosis and ferroptosis, support of regeneration, prevention of lipotoxicity and ER stress, promotion of antioxidant effects, and donation of mitochondria. The underlying mechanisms suggest very close interdependence, including direct cytoplasm and organelle exchange. However, despite the abundance of published papers, the mechanisms of mutual communication between MSCs and hepatocytes are still not fully elucidated.

With regard to parenchyma-stroma crosstalk, the liver is not an exception. MSCs and other stromal cells are present and participate in the arrangement of the parenchymal cell niches in virtually all tissues and organs. It is quite probable that the ways they operate at different locations are similar and that stromal cells in various tissues give a coordinated response to physiological and harmful stimuli. Then, it is time to acknowledge the existence of the fourth system (after the nervous, endocrine, and immune systems) supporting the integrated reactions of the organism. This system can utilize the resident and blood-born MSCs of the bone marrow, adipose tissue, or tissues of another origin for further and deeper integration.

## Figures and Tables

**Figure 1 ijms-24-15212-f001:**
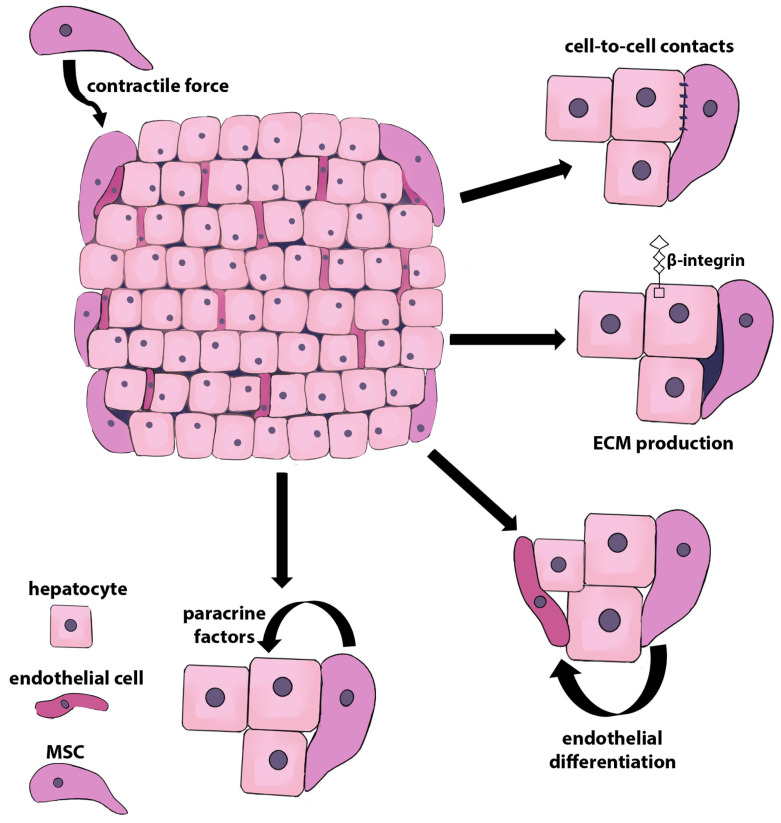
The role of MSCs in the formation of liver organoids. MSCs provide the contractile force for the formation of the heterotypic hepatic organoids and enhance the viability of hepatocytes either through direct contact between hepatocytes and MSCs or via paracrine factors contributing to the proliferation and maintenance of the functional activity of hepatocytes. MSCs produce ECM proteins, facilitating the adhesion of hepatocytes, and enhance the expression of β-integrin by hepatocytes, contributing to the transition of hepatocytes to the S phase of the cell cycle. Within the organoids, MSCs are able to differentiate into endotheliocytes, which form structures similar to a capillary network.

**Figure 2 ijms-24-15212-f002:**
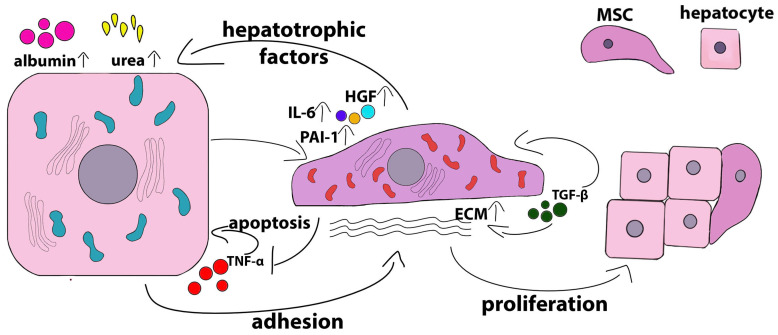
Reciprocal effects of hepatocytes and MSCs in direct co-cultures. MSCs produce hepatotrophic factors, including HGF, IL-6, and PAI-1, while their expression, in turn, is enhanced by hepatocytes. Hepatotrophic factors enhance the survival and proliferation of hepatocytes and improve their functional activity, as witnessed by heightened albumin production and ureagenesis. MSCs secrete TGF-β, which, through autocrine regulation, stimulates the production of ECM proteins, thereby facilitating hepatocyte adhesion and promoting their proliferation. On the other hand, TNF-α secreted by primary hepatocytes can induce their own apoptosis also via autocrine mechanisms. MSCs significantly reduce the expression of TNF-α by hepatocytes, thereby protecting hepatocytes from cell death. HGF—hepatocyte growth factor; IL-6—interleukin 6; PAI-1—plasminogen activator inhibitor-1; TNF-α—tumor necrosis factor α; TGF-β—trans-forming growth factor β; ECM—extracellular matrix.

**Figure 3 ijms-24-15212-f003:**
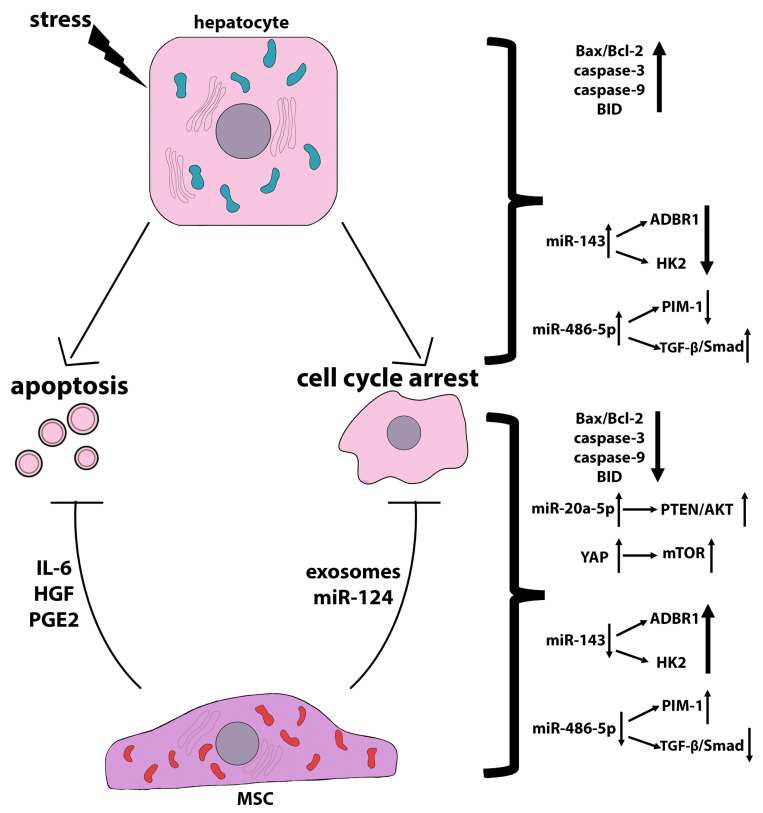
Mechanisms of the anti-apoptotic action of MSCs in the hepatocyte–MSC co-cultures. MSCs inhibit stress-induced apoptosis and cell cycle arrest in hepatocytes through the secretion of the hepatotrophic factors IL-6, HGF, and PGE2, as well as through the production of the miRNA-loaded exosomes. In stressed hepatocytes, MSCs downregulate the expression of the proapoptotic proteins Bax, cleaved caspase 3, and BID and upregulate the expression of the anti-apoptotic protein Bcl-2. MSC-produced PGE2 activates two interconnected signaling pathways involved in liver regeneration. YAP activation is mediated through p-CREB and leads to the suppression of PTEN by miR-29a-3p and subsequent activation of mTOR signaling, ultimately leading to the inhibition of apoptosis and an increase in hepatocyte proliferation. MSCs reduce the miR143 level in hepatocytes subjected to oxidative stress and, accordingly, increase the levels of HK2 and ADRB1. This results in decreased oxidative phosphorylation and a switch to glycolysis, ultimately leading to hepatocyte proliferation and protection from apoptosis. MSCs attenuate damage caused by oxidative stress in hepatocytes by inhibiting miR-486-5p, upregulating PIM1, and blocking TGF-β/Smad signaling. MSCs activate PTEN/AKT signaling pathway in damaged hepatocytes by upregulating miR-20a-5p, thus enhancing their proliferation. MSC-derived exosomal miR-124 promoted liver regeneration by enhancing hepatocyte proliferation through the downregulation of Foxg1. IL-6—interleukin-6; HGF—hepatocyte growth factor; PGE2—prostaglandin E2; YAP—Yes-associated protein; p-CREB—p-cAMP-responsive element binding protein; PTEN—phosphatase and tensin homolog; mTOR—mammalian target of rapamycin; HK2—hexokinase 2; ADRB1—beta-1-adrenergic receptor; PIM1—proviral integration site for Moloney murine leukemia virus kinase 1; TGF-β—transforming growth factor-β.

**Figure 4 ijms-24-15212-f004:**
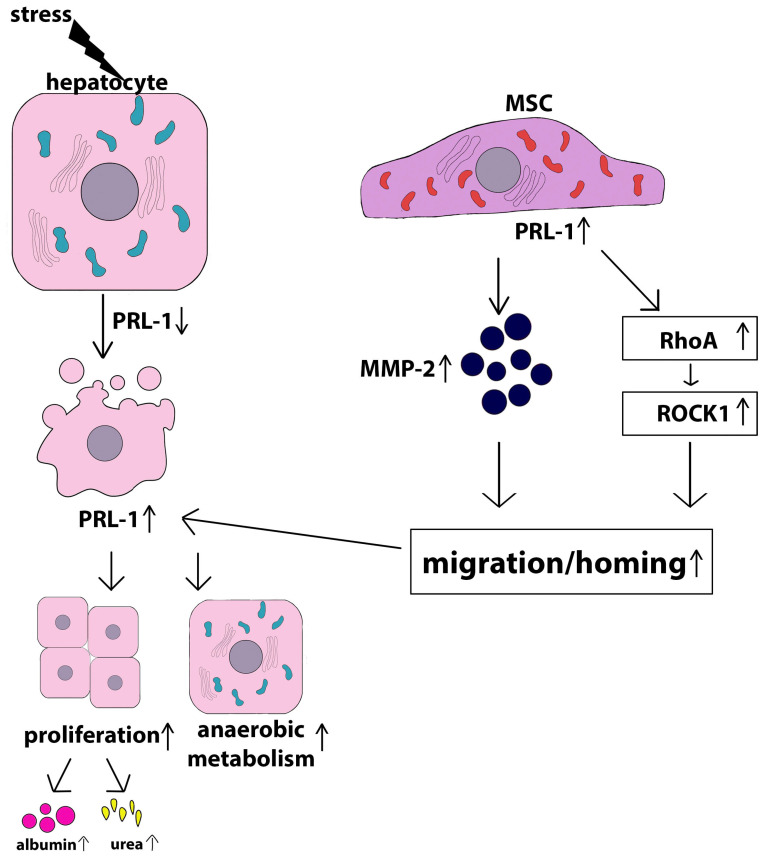
The PRL-1-dependent mechanism of hepatoprotective action of MSCs. Normal hepatocytes and MSCs both express PRL-1. After treatment of primary rat hepatocytes with lithocholic acid, the level of PRL-1 expression decreased. Co-cultivation with MSCs resulted in the restoration of PRL-1 expression in hepatocytes. MSC homing was regulated by RhoA-mediated ROCK1 signaling. PRL-1 produced by hepatocytes acted as a chemokine for MSCs due to increased MMP-2 expression in MSCs. On the other side, PRL-1 produced by MSCs increased anaerobic mitochondrial metabolism in damaged hepatocytes, decreasing cytoplasmic lactate and increasing mitochondrial lactate, which ultimately led to increased ATP synthesis and hepatocyte repair. PRL-1—phosphatase of regenerating liver; MMP-2—matrix metalloproteinase 2; ATP—adenosine triphosphate; RhoA—Ras homolog family member A; ROCK1—Rho-associated coiled-coil-containing protein kinase 1.

**Figure 5 ijms-24-15212-f005:**
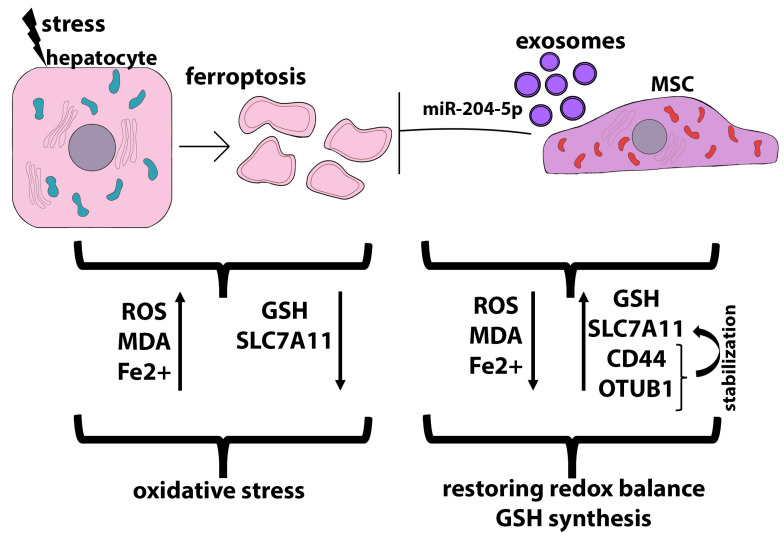
Hepatocyte protection from ferroptosis mediated by MSC or MSC derivatives. MSC-derived exosomes loaded with miR-204-5p inhibit ferroptosis by reducing Fe^2+^ intracellular overload and ROS and MDA levels and by upregulating the expression of GPX4, a protective enzyme that prevents ferroptosis. Also, MSC-derived exosomes induce the stabilization of SLC7A11 via increased expression of CD44 and OTUB1. Thus, MSC may prevent hepatocyte ferroptosis by activation of the antioxidant system.

**Figure 6 ijms-24-15212-f006:**
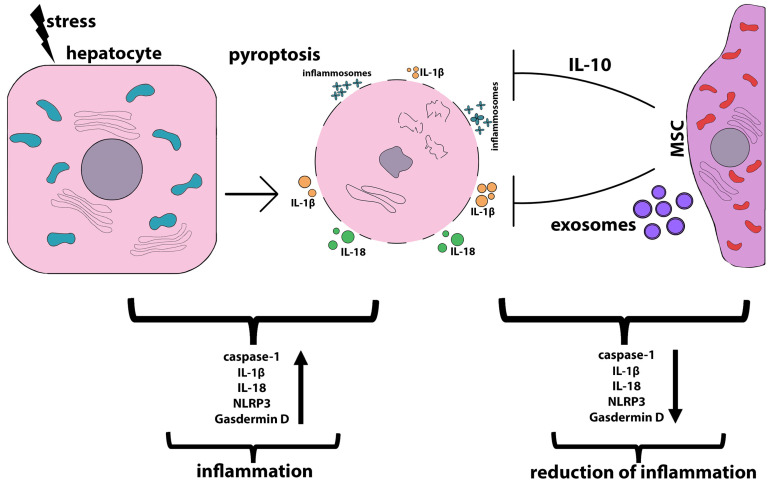
Hepatocyte protection from pyroptosis by MSC or MSC derivatives. MSCs reduce NLRP3-caspase 1 inflammasomes in stressed hepatocytes via the expression of the IL-10 anti-inflammatory cytokine. Also, MSC-derived exosomes reduce the expression of the markers of pyroptosis NLRP3, GSDMD, caspase 1, IL-1β, and IL-18 and the suppression of hepatocyte pyroptosis, probably due to the reduction in inflammation. NLRP3—NLR family pyrin domain containing 3; IL-10—interleukin-10; GSDMD—gasdermin D; IL-1β—interleukin-1β; IL-18—interleukin-18.

**Figure 7 ijms-24-15212-f007:**
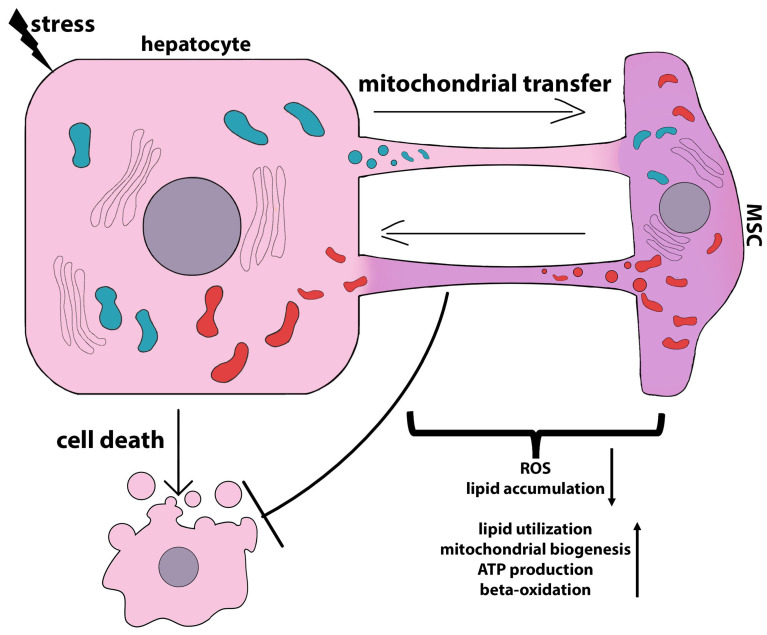
Reciprocal mitochondrial transfer between stressed hepatocytes and MSCs. MSCs prevent hepatocyte cell death by donating their mitochondria to hepatocytes. This process promotes the recovery of lipid utilization, averts lipid accumulation, stimulates the biogenesis of mitochondria, and increases ATP production.

**Table 1 ijms-24-15212-t001:** Bioactive molecules and EVs protecting hepatocytes from apoptosis produced by MSCs.

MSC Tissue Source	Culture Setup and/or Bioactive Agents	Hepatocytes	Anti-Apoptotic Effects and References
BM-MSCs	Exosomes	Primary rat hepatocytes treated with D-galactosamine and lipopolysaccharide	Increase in autophagosome numbers; increase in the expression levels of the autophagy-related proteins LC3II and Beclin-1 and the anti-apoptotic protein Bcl-2; decrease in proapoptotic proteins Bax and cleaved caspase 3 [92]
hESC-derived HuES9.E1 MSCs	Exosomes	TAMH, an immortalized mouse hepatocyte cell line derived from transgenic MT42 male mice overexpressing TGF-α; THLE-2, an immortalized primary human hepatocyte cell line that expresses phenotypic characteristics of normal adult liver epithelial cells; HuH-7, a well-differentiated human hepatocarcinoma cell line; APAP or H_2_O_2_ treatment	Transition from the G0 to G1 phase of the cell cycle; upregulation of TNF-α, IL-6, iNOS, COX-2, and MIP-2; restoration of NF-κB and STAT3 signaling; inhibition of caspase 3; upregulation of anti-apoptotic protein Bcl-xL [93]
Rat BM-MSCs	Indirect co-culture or BM-MSC-CM	Rat hepatocytes	Increase in albumin secretion and urea production [99]
Human orbital fat-derived stem cells	Direct or indirect co-culture IL-6	Rat hepatocytes treated with serum from an acute liver failure patient	Increase in cell viability; increase in albumin secretion and urea production [100]
Human umbilical cord MSCs	Indirect co-culture HGF, EGF, and IL-6	Human LO2 cell line treated with acetaminophen	Protection from APAP-induced necrosis; increase in cell viability [110]
Mouse BM-MSCs	MSC-CM; prostaglandin E2	LPS-treated mouse hepatocyte cell line AML12	Apoptosis inhibition; increase in cell proliferation; YAP activation, suppression of PTEN due to miR-29a-3p, and activation of mTOR signaling [91]

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
