# Peer review of "The Crosstalk between Mesenchymal Stromal/Stem Cells and Hepatocytes in Homeostasis and under Stress"

_ijms, 2023, doi:10.3390/ijms242015212_

Round 1

Reviewer 1 Report

The Review entitled “The crosstalk between mesenchymal stromal/stem cells and hepatocytes in homeostasis and under stress” by Irina V. Kholodenko, Roman V. Kholodenko , Konstantin N. Yarygin describes the key factors mediating the cross-talk between MSCs and hepatocytes, and determines the possible mechanisms of interaction of the two cell types under normal and stressful conditions.

 The review is well written and with a good iconographic heritage; however, it could be improved.

In the introduction, the authors could deepen the specific characteristics of MSCs of hepatic origin compared to MSCs isolated from other body districts.

Are Liver MSCs able to transdifferentiate?

Does this pool of stem cells retain or lose its characteristics during inflammatory pathological conditions or cancer? For example, MSCs isolated from nasal polyps lose not differentiate towards some lineages compared to cells isolated from healthy tissue.

Cho JS, Park JH, Kang JH, Kim SE, Park IH, Lee HM. Isolation and characterization of multipotent mesenchymal stem cells in nasal polyps. Exp Biol Med (Maywood). 2015 Feb;240(2):185-93. doi: 10.1177/1535370214553898. Epub 2014 Oct 6. PMID: 25294891; PMCID: PMC4935324.

Chiarella E, Lombardo N, Lobello N, Piazzetta GL, Morrone HL, Mesuraca M, Bond HM. Deficit in Adipose Differentiation in Mesenchymal Stem Cells Derived from Chronic Rhinosinusitis Nasal Polyps Compared to Nasal Mucosal Tissue. Int J Mol Sci. 2020 Dec 3;21(23):9214. doi: 10.3390/ijms21239214. PMID: 33287173; PMCID: PMC7730671.

Could the authors better explain what metabolic changes arise in hepatoblasts when co-cultured with MSCs?

The section about the conditioned media is very interesting. The authors could deepen this aspect by talking about the impact of conditioned media derived from MSCs treated with drugs and / or phototherapeutics, on hepatocytes.

 Could the authors summarize paragraph 3.1.1 with a table?

Authors can make a cartoon describing the Il-6/ IL6R mechanism of action. Are there competitive IL-6 agonists that can stimulate the pathway? The authors could elaborate on this aspect.

In 3.2 the authors do not elaborate on the role of PRL-2/3.

The MSCs protection from lypotoxicity and ER stress is supported by metabolomic studies?

Minor editing of English language required

Author Response

Dear Reviewer,

Thank you very much for reviewing our manuscript (manuscript ID: ijms-2654787) and your most considerate and helpful comments and suggestions. We revised the manuscript accordingly and feel that it became better.

The Review entitled “The crosstalk between mesenchymal stromal/stem cells and hepatocytes in homeostasis and under stress” by Irina V. Kholodenko, Roman V. Kholodenko , Konstantin N. Yarygin describes the key factors mediating the cross-talk between MSCs and hepatocytes, and determines the possible mechanisms of interaction of the two cell types under normal and stressful conditions.

 The review is well written and with a good iconographic heritage; however, it could be improved.

In the introduction, the authors could deepen the specific characteristics of MSCs of hepatic origin compared to MSCs isolated from other body districts.

  • We briefly describe the properties of the MSCs isolated from liver tissue in comparison with MSCs obtained from other tissue sources in the Introduction (Lines 77-80). It should be noted, though, that these “liver MSCs” may be either resident liver cells or blood born MSCs originating from bone marrow.

Are Liver MSCs able to transdifferentiate?

  • MSCs isolated from the liver are able to differentiate into hepatocyte-like cells to a greater extent than umbilical cord MSCs. We described this in the Introduction (Lines 77-78).

Does this pool of stem cells retain or lose its characteristics during inflammatory pathological conditions or cancer? For example, MSCs isolated from nasal polyps lose not differentiate towards some lineages compared to cells isolated from healthy tissue.

Cho JS, Park JH, Kang JH, Kim SE, Park IH, Lee HM. Isolation and characterization of multipotent mesenchymal stem cells in nasal polyps. Exp Biol Med (Maywood). 2015 Feb;240(2):185-93. doi: 10.1177/1535370214553898. Epub 2014 Oct 6. PMID: 25294891; PMCID: PMC4935324.

Chiarella E, Lombardo N, Lobello N, Piazzetta GL, Morrone HL, Mesuraca M, Bond HM. Deficit in Adipose Differentiation in Mesenchymal Stem Cells Derived from Chronic Rhinosinusitis Nasal Polyps Compared to Nasal Mucosal Tissue. Int J Mol Sci. 2020 Dec 3;21(23):9214. doi: 10.3390/ijms21239214. PMID: 33287173; PMCID: PMC7730671.

  • As shown by Raicevic et al. [Raicevic G, Najar M, Najimi M, El Taghdouini A, van Grunsven LA, Sokal E, Toungouz M. Influence of inflammation on the immunological profile of adult-derived human liver mesenchymal stromal cells and stellate cells. Cytotherapy. 2015 Feb;17(2):174-85. doi: 10.1016/j.jcyt.2014.10.001.], inflammatory conditions significantly influence the immunophenotype of liver MSCs, as stated in the Introduction (Lines 80-85). According to our data, the inflammatory cytokine TNF-α did not affect the increase in the sensitivity of liver MSCs to Fas-induced apoptosis, to which they turned out to be highly resistant [Kholodenko IV, Gisina AM, Manukyan GV, Majouga AG, Svirshchevskaya EV, Kholodenko RV, Yarygin KN. Resistance of Human Liver Mesenchymal Stem Cells to FAS-Induced Cell Death. Curr Issues Mol Biol. 2022 Jul 30;44(8):3428-3443. doi: 10.3390/cimb44080236.]

Could the authors better explain what metabolic changes arise in hepatoblasts when co-cultured with MSCs?

  • In section 2.1 we added a paragraph in which we described how MSCs affect hepatoblasts when co-cultured.

The section about the conditioned media is very interesting. The authors could deepen this aspect by talking about the impact of conditioned media derived from MSCs treated with drugs and / or phototherapeutics, on hepatocytes.

  • We have added a paragraph at the end of section 3.4.

Could the authors summarize paragraph 3.1.1 with a table?

  • We have made a table for paragraph 3.1.1.

Authors can make a cartoon describing the Il-6/ IL6R mechanism of action. Are there competitive IL-6 agonists that can stimulate the pathway? The authors could elaborate on this aspect.

  • Almost 30 years ago, a collection of human IL-6 mutants with receptor agonistic properties was obtained [Toniatti C, Cabibbo A, Sporena E, Salvati AL, Cerretani M, Serafini S, Lahm A, Cortese R, Ciliberto G. Engineering human interleukin-6 to obtain variants with strongly enhanced bioactivity. EMBO J. 1996 Jun 3;15(11):2726-37.]. Among them, D-6 is a high affinity IL-6 variant. D-6 binds to IL-6R ∼40-fold tighter than wild-type IL-6 and exhibits much greater biological activity compared to wild-type IL-6, especially when used in conjunction with sIL-6R on cells expressing only gp130. For example, D-6 alone did not induce responses in endothelial cells, but in combination with sIL-6R caused significantly higher production of MCP-1 compared to IL-6 itself [Romano M, Sironi M, Toniatti C, Polentarutti N, Fruscella P, Ghezzi P, Faggioni R, Luini W, van Hinsbergh V, Sozzani S, Bussolino F, Poli V, Ciliberto G, Mantovani A. Role of IL-6 and its soluble receptor in induction of chemokines and leukocyte recruitment. Immunity. 1997 Mar;6(3):315-25. doi: 10.1016/s1074-7613(00)80334-9.]. Therapeutic approaches targeting IL-6 signaling can be divided into three categories: direct blockade of IL-6, targeting a receptor such as IL-6R or gp130, or targeting of downstream kinase or transcription factors in the JAK-STAT (Janus kinase signal transducer and activator of transcription) pathway. The first two approaches use therapeutic monoclonal antibodies, including sirukumab and siltuximab directly target the IL-6 ligand and block classic signaling (through the membrane-bound IL-6R) and trans-signaling (through the sIL6R) but not IL-6 trans-presentation, tocilizumab and sarilumab that are directed against the IL-6R block all 3 kinds of IL-6 signaling. The third approach uses various selective small molecule inhibitors.JAK inhibitors such as tofacitinib and baricitinib impact upon downstream intracellular signaling [Ridker PM, Rane M. Interleukin-6 Signaling and Anti-Interleukin-6 Therapeutics in Cardiovascular Disease. Circ Res. 2021 May 28;128(11):1728-1746. doi: 10.1161/CIRCRESAHA.121.319077.]. Since these approaches are not currently used for the treatment of liver diseases, we chose not to include their description in the text of the article.

In 3.2 the authors do not elaborate on the role of PRL-2/3.

  • We have added the description of the PRL-2/3 functions in section 3.2 (Lines 758-764 and 769-770)

The MSCs protection from lypotoxicity and ER stress is supported by metabolomic studies?

  • We have added a paragraph to section 3.4 describing metabolomics studies (Lines 997-1007)

Yours truly,

I.V. Kholodenko

Reviewer 2 Report

This review described the hepatocyte-MSC interaction relating synthesis of the supportive extracellular matrix proteins, prevention of apoptosis, pyroptosis and ferroptosis, support of regeneration, and donation of mitochondria etc. It was enjoyable to read all of them. However, it was likely also a highly speculative based on an in vitro co-culture system. Thus, it was likely less of an interest as the therapeutic, medical, and physiological background.

Comments:

1.         Introduction is basically very interesting, except for the final sentence. In the therapeutic view point, in vivo experiments and confirmation are important and inevitable. Therefore, the context should be composed clearly, divided into in vitro and in vivo data (evidence).

2.         It seems that the context mainly consists of in vitro data, so a section specifying in vivo data may be helpful to the readers (both if exist or not).

3.         About the Figure 1, the authors used the term 'trans differentiation'. Is this true? The MSC is considered the stem cells, thus, 'differentiation into endothelial cells' may be correct. Actually, the capacity of tissue specific stem cell differentiation into endothelial cells has been largely reported. In the biological true sense, 'transdifferentiation' is near-impossible. This can only be done through iPS method, although it is not a real transdifferentiation. However, the authors used this word throughout the text. In this context, what are hepatic MSCs (mesenchymal stem cells)? Is this a cell that has been established or differentiated from stem cells?

4.         Similarly, how does the in vivo evidence of the multi-differentiation capacity of hepatic MSCs following cell transplantation? For example, the trace of using EGFP labeling. I certainly want to see this evidence.

5.         Similarly, could you isolate the hepatic MSCs?

6.         Did the endothelial cells from the hepatic MSCs form the capillaries in the liver tissue?  

7.         After the figure 1, the identification of endothelial cells and MSC picture is confusing and the reader may have to further look at Figure 1. Therefore, insert the identification in each figure 2-7.

8.         Through figure 2-7, it should be clarified that the speculative and evidential parts.

9.         From the conclusion, I suggest that the title should be included the words “based on in vitro culture” and/or “speculative/possible interaction mechanism” or similar kind of meanings.

Re-consider the word "transdifferentiation".

Author Response

Dear Reviewer,

Many thanks for reviewing our manuscript (manuscript ID: ijms-2654787) and most fruitful discussion. We revised the manuscript accordingly and think that it became better.

This review described the hepatocyte-MSC interaction relating synthesis of the supportive extracellular matrix proteins, prevention of apoptosis, pyroptosis and ferroptosis, support of regeneration, and donation of mitochondria etc. It was enjoyable to read all of them. However, it was likely also a highly speculative based on an in vitro co-culture system. Thus, it was likely less of an interest as the therapeutic, medical, and physiological background.

Comments:

  1. Introduction is basically very interesting, except for the final sentence. In the therapeutic view point, in vivo experiments and confirmation are important and inevitable. Therefore, the context should be composed clearly, divided into in vitro and in vivo data (evidence).

- In our article, we placed the main emphasis on the molecular mechanisms of the interaction of two types of cells: hepatocytes and mesenchymal stem cells, in order to establish possible mechanisms and pathways of their communication and intercellular signaling. To do this, we selected literature sources in such a way as to be able to really identify molecular/cellular tools and mechanisms of communication, and that is why the focus was made on the in vitro data. And we wrote about this in the Introduction. We fully realize that the in vivo experiments are much more conclusive for assessing the therapeutic role of MSCs. However, in the experiments utilizing animal models it is almost impossible to identify the communication pathways of two separate cell types, since they interfere with many other cell types (liver cells, blood cells, vascular cells), as well as the extracellular matrix and a huge range of secreted biologically active substances located in the extracellular environment. Actually, we are talking about the dichotomy between reductionistic and holistic science (see, for example, this paper: doi: 10.1128/IAI.01343-10) with molecular biology representing the first, systems biology the latter, and cell biology, physiology, etc. are somewhere in-between. It is important to stress that in fact holistic science is impossible without reductionistic science providing the facts for holistic science to work with. 

  1. It seems that the context mainly consists of in vitro data, so a section specifying in vivo data may be helpful to the readers (both if exist or not).

- As we explained above, the purpose of this review was to identify potential crosstalk pathways between hepatocytes and mesenchymal stem cells. In this regard, in vitro data are presented in more detail and extensively, while in vivo results are selected in such a way as to demonstrate that in animal models of liver diseases the therapeutic effects of MSCs and their derivatives are really manifested, while the mechanisms of these therapeutic effects are, for the most part, elucidated in the in vitro experiments.

  1. About the Figure 1, the authors used the term 'trans differentiation'. Is this true? The MSC is considered the stem cells, thus, 'differentiation into endothelial cells' may be correct. Actually, the capacity of tissue specific stem cell differentiation into endothelial cells has been largely reported. In the biological true sense, 'transdifferentiation' is near-impossible. This can only be done through iPS method, although it is not a real transdifferentiation. However, the authors used this word throughout the text. In this context, what are hepatic MSCs (mesenchymal stem cells)? Is this a cell that has been established or differentiated from stem cells?

- We have replaced the term “transdifferentiation” with “differentiation” throughout the text and in Figure 1.

  1. Similarly, how does the in vivo evidence of the multi-differentiation capacity of hepatic MSCs following cell transplantation? For example, the trace of using EGFP labeling. I certainly want to see this evidence.

- There are no data regarding multilineage differentiation of hepatic MSCs in vivo in the literature. However, there are several studies demonstrating that liver MSCs are capable of differentiating into hepatocytes in vivo [Najimi M., Khuu D.N., Lysy P.A., Jazouli N., Abarca J., Sempoux C., Sokal E.M. Adult-derived human liver mesenchymal-like cells as a potential progenitor reservoir of hepatocytes? Cell Transpl. 2007;16:717–728. doi: 10.3727/000000007783465154.; Khuu D.N., Nyabi O., Maerckx C., Sokal E., Najimi M. Adult human liver mesenchymal stem/progenitor cells participate in mouse liver regeneration after hepatectomy. Cell Transpl. 2013;22:1369–1380. doi: 10.3727/096368912X659853.; Pan Q., Fouraschen S.M., Kaya F.S., Verstegen M.M., Pescatori M., Stubbs A.P., van Ijcken W., van der Sloot A., Smits R., Kwekkeboom J., et al. Mobilization of hepatic mesenchymal stem cells from human liver grafts. Liver Transpl. 2011;17:596–609. doi: 10.1002/lt.22260.]. Also, there are works that show this ability, for example, for bone marrow MSCs [Anjos-Afonso F, Siapati EK, Bonnet D. In vivo contribution of murine mesenchymal stem cells into multiple cell-types under minimal damage conditions. J Cell Sci. 2004 Nov 1;117(Pt 23):5655-64. doi: 10.1242/jcs.01488.]. Given the great similarity and, possibly, common origin between MSCs in different tissue niches, one can suggest that these cells, upon entering a specific tissue, are able to differentiate into different cell types.

  1. Similarly, could you isolate the hepatic MSCs?

- We isolate MSCs from human liver. We have several publications on this topic [DOI: 10.3390/cells8101127; DOI: 10.3390/cimb44080236; DOI: 10.18097/PBMC20166206674].

  1. Did the endothelial cells from the hepatic MSCs form the capillaries in the liver tissue?

- There is no published data on whether capillaries in liver tissue are formed from MSC-derived endothelial cells. However, there are results proving that transplanted MSCs are able to differentiate into endothelial cells in vivo, for example, in a model of myocardial infarction [Silva GV, Litovsky S, Assad JA, Sousa AL, Martin BJ, Vela D, Coulter SC, Lin J, Ober J, Vaughn WK, Branco RV, Oliveira EM, He R, Geng YJ, Willerson JT, Perin EC. Mesenchymal stem cells differentiate into an endothelial phenotype, enhance vascular density, and improve heart function in a canine chronic ischemia model. Circulation. 2005 Jan 18;111(2):150-6. doi: 10.1161/01.CIR.0000151812.86142.45.]. Because studies related to liver pathologies have focused on restoring hepatocytes and their functions, as well as reducing inflammation, issues related to liver neovascularization are, unfortunately, generally not addressed.

  1. After the figure 1, the identification of endothelial cells and MSC picture is confusing and the reader may have to further look at Figure 1. Therefore, insert the identification in each figure 2-7.

- Done.

  1. Through figure 2-7, it should be clarified that the speculative and evidential parts.

- Figures 2-7 present only those data that have been proven in the in vitro experiments and, to a certain degree, confirmed in the in vivo studies.

  1. From the conclusion, I suggest that the title should be included the words “based on in vitro culture” and/or “speculative/possible interaction mechanism” or similar kind of meanings.

- The title of the manuscript has been modified.

Comments on the Quality of English Language

Re-consider the word "transdifferentiation".

  • Done.

Yours truly,

I.V. Kholodenko

Round 2

Reviewer 1 Report

The authors significantly improved thei Review, so it is now publishable on ijms

Minor editing of English language required

Reviewer 2 Report

The standing position of this review article has been cleared now.